# Beyond Extrapolation: Knowledge Utilization Paradigm with Bidirectional Inspiration for Time Series Forecasting

Liu Chong [1]  Yingjie Zhou [* 1]  Hao Li [1]  Pengyang Wang [2]  Qingsong Wen [3]  Ce Zhu [4]

## Abstract

Time-series forecasting is critical in various scenarios, such as energy, transportation, and public health. However, most existing forecasters rely primarily on one-way inference, *i.e.*, mapping **history** to **target**, and overlook the structural information provided by a revised natural chain ("**history** (model input) – **target** (ground-truth output) – **post-target continuation**"). The post-target continuation records how trajectories evolve after the target, which can help stabilize forecasting, but it is not observable at inference time. In this work, we aim to obtain an approximate proxy of the post-target continuation for the current input, providing structural knowledge for bidirectional forecasting. This idea is instantiated as KUP-BI (Knowledge Utilization Paradigm with Bidirectional Inspiration), a new time-series modeling paradigm that distills continuation-style knowledge (as an approximate post-target continuation proxy) from a *train-only* historical library and integrates it into standard forecasting backbones. The input stream and the continuation-proxy stream are fused via a lightweight feature-level gating module. This design does not introduce information beyond what is already contained in the training trajectories; instead, it provides a structured inductive bias that helps backbones exploit typical continuation patterns rather than relying solely on parametric extrapolation. Experimental results on six public datasets show that KUP-BI consistently improves the forecasting performance of state-of-the-art models, with small additional overhead. Our

code is available at `https://github.com/yihannnnn/KUP-BI`.

[1] The College of Computer Science, Sichuan University, Chengdu 610065, China [2]Computer and Information Science, University of Macau [3]Squirrel Ai [4]School of Information and Communication Engineering, University of Electronic Science and Technology of China. Correspondence to: Yingjie Zhou <yjzhou09@gmail.com>.

*Proceedings of the $43^{rd}$ International Conference on Machine Learning*, Seoul, South Korea. PMLR 306, 2026. Copyright 2026 by the author(s).

## 1. Introduction

Time-series forecasting plays an important role in various scenarios, such as finance (Huang et al., 2024), traffic (Zeng et al., 2023b), weather (Lam et al., 2023), and energy (Wang et al., 2019). As the demand for accurate predictions grows, forecasting methods have evolved from single-step to multi-step settings (Zhou et al., 2021; Wu et al., 2021) and from linear (Box et al., 2015) to nonlinear models (Li et al., 2023; Shao et al., 2025). Recent work shows that deep learning models can capture complex nonlinear patterns and improve long-horizon forecasting on real-world data (Kudrat et al., 2025; Liu et al., 2025).

Although recent deep learning forecasters have achieved notable progress, they typically follow the one-way paradigm that predicts the **target** from the **history** (Shao et al., 2025; Wen et al., 2023). Under long prediction horizons, this modeling approach struggles to stably infer the evolution trend within the target window, often leading to error accumulation and trend drift. To alleviate this issue, some recent works attempt to reuse target information from training trajectories as auxiliary information (Han et al., 2025; Tire et al., 2024). However, target-level information is highly aligned with the supervision and can become an overly strong shortcut during training, which may weaken generalization. In contrast, **post-target continuation** shares the same underlying dynamics as the target but is temporally decoupled from the target window, providing a weaker yet more transferable cue about how the system tends to evolve. As a result, they act as a structural prior rather than a direct template. Motivated by this, we revisit the natural chain in the training data, "**history** (model input) – **target** (ground-truth output) – **post-target continuation**", and propose to distill continuation-style auxiliary features from such chains to guide forecasting. This enables a new modeling paradigm that leverages not only the forward mapping from history to target, but also the backward structural cues implied by post-target evolution, resulting in a bidirectional-inspired forecasting framework.

Building on this perspective, we propose the **Knowledge Utilization Paradigm with Bidirectional Inspiration (KUP-BI)**, which augments the standard **history**-to-**target** pathway with a continuation-style auxiliary stream $\mathbf{Z} = f(\mathbf{X}, \mathcal{D})$ constructed from a train-only historical library $\mathcal{D}$, where $\mathbf{Z}$ summarises how training trajectories with histories similar to the current input $\mathbf{X}$ have tended to continue beyond their targets. Specifically, KUP-BI constructs a train-only library from **history–target–post-target continuation** chains extracted from the training set, where each retrieval entry is formed from one such chain and consists of an offset-aligned history together with a ratio-style continuation transformation that describes how the post-target continuation changes relative to its history. Given a new input $\mathbf{X}$, we retrieve similar training histories from this library $\mathcal{D}$ in a channel-wise manner, aggregate their associated ratio-style transformations for each channel via temperature-controlled softmax weighting, and stack the fused channel-wise signals into a transformation matrix, which is then applied to $\mathbf{X}$ to obtain the continuation-style auxiliary input $\mathbf{Z}$. In KUP-BI, features extracted from $\mathbf{X}$ and $\mathbf{Z}$ are fused via a lightweight gating module, enabling the backbone to exploit both the local information in $\mathbf{X}$ and typical continuation patterns distilled from the library $\mathcal{D}$. Conceptually, this provides an additional, data-driven inductive bias that encourages forecasts to align with the kinds of post-target evolution commonly observed in the training data, rather than relying solely on parametric extrapolation from the current history. Figure 1 provides an overview of KUP-BI and highlights how the continuation-style auxiliary stream offers structural guidance beyond one-way history-to-target mapping. *It is important to note that KUP-BI is not bound to any specific mechanism for constructing this continuation-style auxiliary feature (see Subsection 4.3 for an alternative prediction-based construction). The novelty of KUP-BI lies in explicitly introducing and exploiting continuation-style auxiliary features derived from training chains, and in demonstrating that injecting such features as a separate stream can consistently improve forecasting performance across diverse backbones.* Our contributions are as follows:

- We propose a new perspective for time-series forecasting that explicitly leverages the full "**history–target–post-target continuation**" chain in the *training data*. Instead of relying solely on one-way extrapolation from history to target, we introduce continuation-style auxiliary features distilled from a train-only historical library and achieve stable forecasts through bidirectional inspiration.
- We instantiate this idea with KUP-BI, a simple optional framework that can be plugged into standard forecasting backbones.
- We demonstrate the generality of KUP-BI by incor-

porating it as a plug-in module into state-of-the-art backbones on six public datasets. Across these settings, KUP-BI consistently improves forecasting performance with small additional overhead.

**(a). Traditional one-way mapping**

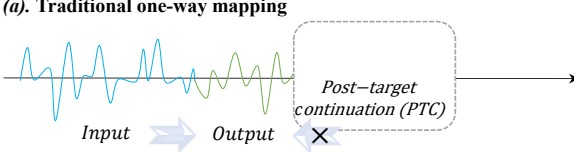

**(b). Knowledge Utilization Paradigm with Bidirectional Inspiration**

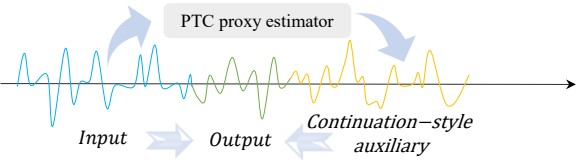

*Figure 1.* Illustration of KUP-BI. (a) Traditional one-way forecasting maps the history (input) to the target (output), as the true post-target continuation (PTC) is unavailable at inference time (marked with ×). (b) KUP-BI introduces a continuation-style auxiliary stream as an approximate proxy of the PTC, constructed from training trajectories via a PTC proxy estimator, and fuses it with the backbone to provide continuation-aware structural guidance and improve long-horizon forecasting. The proxy estimator is instantiated with a retrieval-based construction from a train-only historical library, while an alternative construction strategy is also discussed in Subsection 4.3.

## 2. Related Work

### 2.1. Time Series Forecasting Models

In recent years, time-series forecasting techniques have developed rapidly to address increasingly complex prediction demands (Liu et al., 2024; Dai et al., 2024; Huang et al., 2025). Different neural network architectures show distinct advantages: convolution-based models (Wu et al., 2023; Luo & Wang, 2024) can efficiently capture local temporal features, Transformer-based models (Nie et al., 2023; Zhou et al., 2022) are effective at modeling long-range dependencies, and MLP-based models (Zeng et al., 2023a; Wang et al., 2024) are known for their simplicity and computational efficiency. With further research, large-scale pretrained models (Jin et al., 2024; Niu et al., 2025) have been explored for time-series forecasting, and these approaches demonstrate good performance in zero-shot and few-shot scenarios. Existing time-series forecasting models, despite their architectural diversity, still follow the traditional single-stream paradigm, relying solely on past-to-future mappings. This unidirectional design limits their ability to leverage future continuation streams as external knowledge, and may reduce robustness under distribution shifts (Kim et al., 2022; Liu et al., 2025).

## 2.2. Retrieval enhancement methods for time-series forecasting

In recent years, retrieval augmentation has rapidly emerged in time-series forecasting. The core idea is to build a retrieval library from training data and reuse retrieved relevant segments to assist prediction during training and/or inference. For foundation time-series models (TSFMs), a typical line of work treats the backbone as a black box or keeps it frozen, using retrieved segments as prompts only at inference time, or training only a lightweight fusion module, such as RAF (Tire et al., 2024) and TS-RAG (Ning et al., 2025). In contrast, methods designed for conventional deep forecasting models often explicitly incorporate retrieved information into the training process, enabling the backbone to learn end-to-end how to exploit retrieval signals, *e.g.*, RAFT (Han et al., 2025). Notably, RAFT can also be applied in a plug-in manner by post-hoc adjusting backbone predictions at the final stage, showing its architectural flexibility. Despite their different implementations, these methods largely focus on the first two parts of the natural supervised chain "history (model input)–target (ground-truth output)–post-target continuation": they mainly reuse the target segments associated with similar historical patterns, either as direct compensation signals or through neighbor aggregation. The former may increase the reliance on target-level information from retrieved samples, especially when retrieval is involved during training, which can amplify label-neighbor dependence. The latter, while more implicit, may still bias the prediction toward an averaged neighborhood and dilute sample-specific future variations, making the model less robust under distribution shifts or multi-modal futures. Different from prior work, KUP-BI explicitly brings the third segment into play by constructing a continuation-style auxiliary stream from the post-target continuation in the training data. By leveraging continuation cues rather than directly reusing target fragments, KUP-BI can mitigate label-neighbor dependence and inject additional future-structure information to stabilize and improve forecasting.

## 3. Methodology

**Problem Settings** We consider an input sequence $\mathbf{X} = [\mathbf{x}_0, \ldots, \mathbf{x}_{L-1}] \in \mathbb{R}^{L \times C}$, where $\mathbf{x}_i \in \mathbb{R}^C$, $L$ denotes the length of the look-back window, and $C$ denotes the number of variables (channels). The goal of time-series forecasting is to learn a mapping from historical observations to the ground-truth target sequence, producing predictions $\hat{\mathbf{Y}} = \{\hat{\mathbf{y}}_i\}_{i=0}^{T-1} \in \mathbb{R}^{C \times T}$ that approximate the ground-truth series $\mathbf{Y} = \{\mathbf{y}_i\}_{i=0}^{T-1} \in \mathbb{R}^{C \times T}$, where $T$ is the forecast horizon. In time-series forecasting, MAE and MSE are standard evaluation metrics, with lower values reflecting better predictive accuracy. Their definitions are as follows:

$$\mathcal{L}_{MAE} = \frac{1}{CT} \sum\nolimits_{c=1}^{C} \sum\nolimits_{i=0}^{T-1} |y_{c,i} - \hat{y}_{c,i}|,$$

$$\mathcal{L}_{MSE} = \frac{1}{CT} \sum\nolimits_{c=1}^{C} \sum\nolimits_{i=0}^{T-1} (y_{c,i} - \hat{y}_{c,i})^2.$$

**Method Overview** As illustrated in Figure 2, KUP-BI decouples *continuation construction* from *forecasting*. We first build a train-only retrieval library that summarizes how post-target continuations evolve *relative to* their preceding histories, and then utilize this information to construct a continuation-style auxiliary signal for each input window in train/validation/test. The auxiliary signal is encoded and fused with the current input representations at the feature level, while the forecasting backbone itself remains unchanged and serves as the sole predictor. In the following, we describe each component in detail, starting from the construction of the retrieval library.

**Retrieval Library.** Given a multivariate time series, each training instance naturally contains a **history** segment $\mathbf{H} \in \mathbb{R}^{L \times C}$, its corresponding **target** segment $\mathbf{Y} \in \mathbb{R}^{T \times C}$ and a subsequent **post-target continuation** segment $\mathbf{F} \in \mathbb{R}^{L \times C}$. In our implementation, for each chain we take a history window and extract a post-target continuation window of the same length that follows the target in time. To obtain a simple description of how the continuation relates to its history, we compute a ratio-style representation between $\mathbf{H}$ and $\mathbf{F}$:

$$\mathbf{R} = (\mathbf{F} - \mathbf{H}) \oslash (\mathbf{H} + \epsilon \, \text{sign}(\mathbf{H})), \qquad (1)$$

where $\oslash$ denotes element-wise division, $\text{sign}(\cdot)$ is taken element-wise and $\varepsilon$ is a small stabiliser that prevents numerical instability near zero. This matrix $\mathbf{R}$ can be viewed as a heuristic relative-change descriptor that highlights how the post-target continuation differs from its history (for example, amplitude rescaling, seasonal strengthening or weakening), rather than as an optimal or unique statistical operator.

We deliberately adopt this closed-form ratio, instead of introducing an additional neural summariser over the library, for two reasons. First, it keeps the continuation construction strictly non-parametric and decoupled from the backbone, so that the same library can be reused across different backbones (or even non-neural forecasters) without retraining. Second, it avoids giving KUP-BI extra trainable capacity compared to the baselines, making it easier to attribute performance gains to how continuation-style information is utilised rather than to a larger model. More flexible learnable encoders over the library are orthogonal extensions and are left to future work.

We build a retrieval library from the **training set** as a collection of offset-anchored history–ratio pairs:

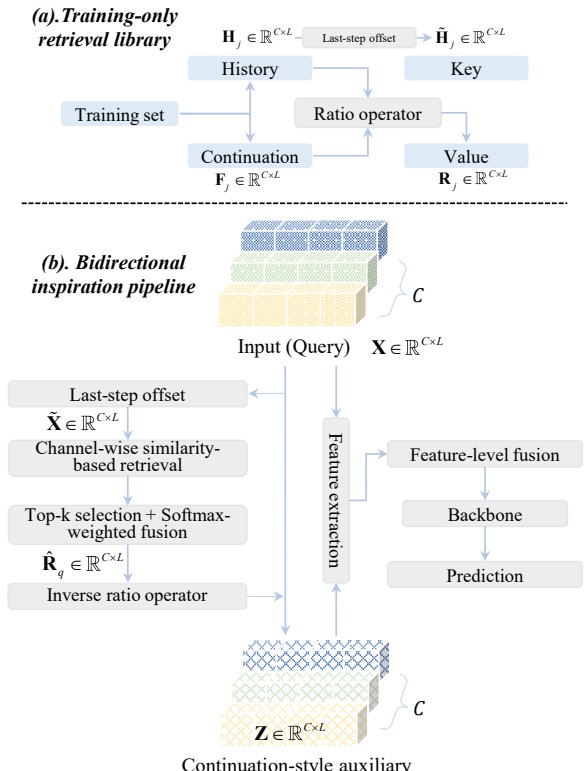

*Figure 2.* **Overview of KUP-BI. (a) Training-only retrieval library.** Each training trajectory is decomposed into a natural chain $(\mathbf{H}_j, \mathbf{Y}_j, \mathbf{F}_j)$ (history, target, and post-target continuation). We convert it into a retrieval entry $(\tilde{\mathbf{H}}_j, \mathbf{R}_j)$, where the key $\tilde{\mathbf{H}}_j$ is obtained from the history via last-step offsetting (Han et al., 2025), and the value $\mathbf{R}_j$ is produced by applying a ratio operator to $(\mathbf{H}_j, \mathbf{F}_j)$, encoding continuation changes relative to the preceding dynamics. Targets $\mathbf{Y}_j$ are **not** reused during retrieval. **(b) Bidirectional inspiration pipeline.** Given a query window $\mathbf{X}$, we apply the same last-step offsetting to obtain $\tilde{\mathbf{X}}$, retrieve similar keys from the constructed library in a channel-wise manner, and aggregate the associated values through Top-$k$ selection and softmax weighting to obtain $\hat{\mathbf{R}}_q$. The resulting continuation-style auxiliary $\mathbf{Z}$ (a proxy for the post-target continuation) is then fused with the input stream via lightweight feature-level gating before being fed into the backbone for prediction.

$$\mathcal{D} = \{(\tilde{\mathbf{H}}_j, \mathbf{R}_j)\}_{j=1}^{N}, \qquad (2)$$

where each entry corresponds to single training chain. Here $\tilde{\mathbf{H}}_j$ denotes the last-step offset version of $\mathbf{H}_j$ (stored for efficiency): $\tilde{\mathbf{H}}_j[t,:] = \mathbf{H}_j[t,:] - \mathbf{H}_j[L,:], \quad t = 1, \ldots, L.$

**Similarity search (Key) via correlation-based candidate selection (channel-wise, offset-anchored).**

Given a query window (current input) $\mathbf{X}_q \in \mathbb{R}^{L \times C}$, we first apply last-step offsetting (Han et al., 2025) to remove local level differences: $\tilde{\mathbf{X}}_q[t,:] = \mathbf{X}_q[t,:] - \mathbf{X}_q[L,:] \quad t = 1, \ldots, L$, where $\mathbf{X}_q[L,:]$ denotes the last time step of the window. We then compute *channel-wise* Pearson correla-

tions (Benesty et al., 2009) between $\tilde{\mathbf{X}}_q^{(:,c)}$ and all $\tilde{\mathbf{H}}_j^{(:,c)}$: $\mathrm{corr}_{j,c} = \mathrm{Corr}(\tilde{\mathbf{X}}_q^{(:,c)}, \tilde{\mathbf{H}}_j^{(:,c)}), \quad c = 1, \ldots, C.$ For each channel $c$, we select its Top-$k$ neighbours according to the largest *positive* correlations:

$$\mathcal{K}_c(\mathbf{X}_q) = \mathrm{Top}\text{-}k\big(\{\max(\mathrm{corr}_{j,c}, 0)\}_j\big). \qquad (3)$$

**Ratio aggregation: softmax-weighted fusion and robust clipping (per channel).** For each channel $c$, we aggregate the corresponding columns of the ratio matrices from its Top-$k$ candidates $\mathcal{K}_c(\mathbf{X}_q)$ using a temperature-controlled softmax over the (positive) correlations:

$$\hat{\mathbf{r}}^{(c)} = \sum_{j \in \mathcal{K}_c(\mathbf{X}_q)} \alpha_{j,c} \mathbf{R}_j^{(:,c)}, \qquad (4)$$

where $\alpha_{j,c} = \dfrac{\exp\big((\mathrm{corr}_{j,c} - m_c)/\tau\big)}{\sum_{\ell \in \mathcal{K}_c(\mathbf{x}_q)} \exp\big((\mathrm{corr}_{\ell,c} - m_c)/\tau\big)}$, $\tau$ is a temperature parameter and $m_c = \max_{\ell \in \mathcal{K}_c(\mathbf{x}_q)} \mathrm{corr}_{\ell,c}$ is subtracted for numerical stability.

Stacking $\{\hat{\mathbf{r}}^{(c)}\}_{c=1}^{C}$ yields the fused ratio matrix $\hat{\mathbf{R}}_q \in \mathbb{R}^{L \times C}$. We then apply quantile–$\tanh$ clipping with the 90th percentile $R'_q = \mathcal{Q}_{0.9}(|\hat{\mathbf{R}}_q|)$:

$$\tilde{\mathbf{R}}_q = R'_q \cdot \tanh\big(\hat{\mathbf{R}}_q / R'_q\big),$$

where the division and $\tanh$ are applied element-wise. This softmax-weighted fusion emphasises candidates that are more strongly correlated with the query while retaining contributions from multiple neighbours, and the quantile-based $\tanh$ clipping provides a simple way to limit extreme ratio values and improve robustness across datasets.

**Continuation-style auxiliary generation.** We apply $\tilde{\mathbf{R}}_q$ to the current input to obtain a continuation-style auxiliary sequence:

$$\hat{\mathbf{F}}_q = \mathbf{X}_q + (\tilde{\mathbf{R}}_q + \epsilon_s \,\mathrm{sign}(\tilde{\mathbf{R}}_q)) \odot \mathbf{X}_q,$$

where $\odot$ denotes element-wise multiplication and $\epsilon_s$ is a very small signed numerical stabilizer introduced for robustness.

To roughly align its scale with the historical stream, we further normalise $\hat{\mathbf{F}}_q$ to match the mean and (per-channel) standard deviation of $\mathbf{X}_q$:

$$\mathbf{Z} = (\hat{\mathbf{F}}_q - \boldsymbol{\mu}_{\hat{\mathbf{F}}_q}) \oslash (\boldsymbol{\sigma}_{\hat{\mathbf{F}}_q} + \varepsilon) \odot (\boldsymbol{\sigma}_{\mathbf{X}_q} + \varepsilon) + \boldsymbol{\mu}_{\mathbf{X}_q}, \quad (5)$$

where $\oslash$ denotes element-wise division, and $\boldsymbol{\mu}_{(\cdot)}$, $\boldsymbol{\sigma}_{(\cdot)}$ are the channel-wise mean and standard deviation computed over the time dimension.

This modulation step is loosely consistent with the ratio definition in (1): it uses the estimated ratio as a feature-wise

scaling of the current history, but we do not treat it as an exact algebraic inverse of (1). Rather, the modulation and normalisation together form a simple practical heuristic that injects continuation-style information into the main stream while keeping the auxiliary branch on a comparable scale.

**Gated fusion.** Let the intermediate features of the historical stream and the continuation-style auxiliary stream be $\mathbf{X}_{\text{main}} = \text{Fea}(\mathbf{X}_q)$ and $\mathbf{X}_{\text{aux}} = \text{Fea}(\mathbf{Z})$, respectively. We compute fusion weights $\gamma$ using either a static learnable gate or an optional lightweight data-dependent gate:

$$\gamma = \begin{cases} \sigma(g), & \text{static gate,} \\ \sigma\big(\phi([\bar{\mathbf{X}}_{\text{main}}, \bar{\mathbf{X}}_{\text{aux}}])\big), & \text{dynamic gate,} \end{cases} \quad (6)$$

where $\bar{\mathbf{X}}_{main}$ and $\bar{\mathbf{X}}_{aux}$ denote summary statistics of the two streams obtained by averaging along the primary structural axis of the corresponding intermediate representation (e.g., temporal axis for sequence features or token/patch axis for patchified features), and $\phi(\cdot)$ is a lightweight shared mapping. The resulting $\gamma$ is broadcast to match the shape of the corresponding intermediate features.

The two streams are first combined via gated fusion:

$$\widetilde{\mathbf{X}} = \gamma \odot \mathbf{X}_{\text{main}} + (1 - \gamma) \odot \mathbf{X}_{\text{aux}}, \quad (7)$$

where $\odot$ denotes element-wise multiplication and the same $\gamma$ is broadcast along the time dimension. To further stabilise training and preserve the dominant role of the main (historical) stream, we apply a controlled residual fusion governed by a coefficient $\alpha \in [0, 1]$: $\mathbf{X}' = \alpha \mathbf{X}_{\text{main}} + (1 - \alpha) \widetilde{\mathbf{X}}$.

This design keeps the final representation as a convex combination of the historical and auxiliary streams, while allowing the exact gate parameterization to adapt to different backbone representations.

**Complexity Analysis.** The proposed KUP-BI comprises three components. First, a correlation-based retrieval stage builds a train-only library and computes correlations offline, caching per-channel candidate lists so this step does not affect training or inference latency. Second, an online continuation-style auxiliary module re-ranks the cached candidates, applies temperature-scaled softmax weighting, and aggregates their ratio signals to form an auxiliary sequence; it then applies quantile-based clipping followed by tanh squashing, together with distribution alignment. Third, a lightweight gated fusion combines the historical stream and the auxiliary stream. Compared to typical backbones (*e.g.*, multi-head attention), these online operations are negligible in runtime and memory.

## 4. Experiments

### 4.1. Experiment Setting

**Datasets** We conduct experiments on six widely used public benchmarks covering electricity, healthcare, and economics: the ETT family (ETTh1, ETTh2, ETTm1, ETTm2) (Zhou et al., 2021), ILI[1], and Exchange Rate (Lai et al., 2018). Detailed dataset statistics and preprocessing procedures are provided in Appendix A.

**Backbones** To evaluate the effectiveness and generality of the proposed KUP-BI, we instantiate it on four strong forecasting backbones with diverse architectures: the Transformer-based PatchTST (Nie et al., 2023), the MLP-based DLinear (Zeng et al., 2023a), the CNN-based TimesNet (Wu et al., 2023), and the hybrid dual-stream (MLP+CNN) xPatch (Stitsyuk & Choi, 2025). Backbone-specific details are summarized in Appendix B.

**Implementation Details** In our experiments, we use the official implementations released by the authors of each backbone method. For fair comparison, when KUP-BI is used to enhance a backbone forecaster, we consider two hyperparameter-tuning protocols: (1) Plugin-only tuning, where we keep the backbone's official hyperparameter setting fixed and tune only the KUP-BI–specific hyperparameters; and (2) Light joint tuning, where we tune KUP-BI together with a small set of the backbone's training hyperparameters (*e.g.*, learning rate) within a limited range. We adopt this design because introducing KUP-BI adds an auxiliary information stream that changes the model's input form and optimization dynamics, meaning that the backbone's official hyperparameters may not be optimal under this augmented setting. By reporting both (1) and (2), we provide a clean attribution of the gains while also reflecting the method's performance upper bound under reasonable tuning. All experiments are implemented in PyTorch (Paszke et al., 2019) and run on 3 NVIDIA RTX 4080 (16 GB each). Additional experimental details, such as where the original input and the auxiliary input are fused, are provided in Appendix C.

### 4.2. Main Results

As shown in Table 1, KUP-BI consistently improves performance across most dataset–backbone combinations, indicating strong cross-architecture generalization and demonstrating that it can serve as a lightweight plug-in to effectively enhance the forecasting accuracy of existing models, with small additional overhead (see Appendix D for runtime analysis). A further comparison between the two configurations shows that the plugin-only setting (tuning only KUP-BI hyperparameters) already yields stable gains,

---

[1]https://gis.cdc.gov/grasp/fluview/fluportaldashboard.html

*Table 1.* Long-term multivariate forecasting results. Results are averaged from all prediction lengths. The best and second-best results are highlighted in red and blue, respectively. Full results are listed in **Appendix F.1**.

| Model | DLinear | | | | | | PatchTST | | | | | | TimesNet | | | | | | xPatch | | | | | |
|---|---|---|---|---|---|---|---|---|---|---|---|---|---|---|---|---|---|---|---|---|---|---|---|---|
| | Ori | | + KUP-BI (Plugin-only) | | + KUP-BI (Joint-tune) | | Ori | | + KUP-BI (Plugin-only) | | + KUP-BI (Joint-tune) | | Ori | | + KUP-BI (Plugin-only) | | + KUP-BI (Joint-tune) | | Ori | | + KUP-BI (Plugin-only) | | + KUP-BI (Joint-tune) | |
| Metric | MSE | MAE | MSE | MAE | MSE | MAE | MSE | MAE | MSE | MAE | MSE | MAE | MSE | MAE | MSE | MAE | MSE | MAE | MSE | MAE | MSE | MAE | MSE | MAE |
| ETTh1 | 0.445 | 0.454 | 0.439 | 0.450 | 0.425 | 0.437 | 0.419 | 0.432 | 0.413 | 0.428 | 0.409 | 0.425 | 0.472 | 0.463 | 0.463 | 0.458 | 0.458 | 0.455 | 0.444 | 0.438 | 0.431 | 0.438 | 0.409 | 0.422 |
| ETTh2 | 0.469 | 0.463 | 0.453 | 0.458 | 0.394 | 0.426 | 0.330 | 0.379 | 0.329 | 0.379 | 0.326 | 0.378 | 0.415 | 0.426 | 0.402 | 0.419 | 0.401 | 0.418 | 0.342 | 0.383 | 0.340 | 0.382 | 0.339 | 0.382 |
| ETTm1 | 0.359 | 0.381 | 0.359 | 0.380 | 0.358 | 0.380 | 0.353 | 0.382 | 0.352 | 0.382 | 0.350 | 0.379 | 0.415 | 0.418 | 0.407 | 0.415 | 0.405 | 0.414 | 0.352 | 0.372 | 0.352 | 0.372 | 0.350 | 0.372 |
| ETTm2 | 0.283 | 0.345 | 0.281 | 0.345 | 0.266 | 0.330 | 0.258 | 0.315 | 0.257 | 0.315 | 0.255 | 0.314 | 0.296 | 0.333 | 0.292 | 0.332 | 0.288 | 0.329 | 0.252 | 0.308 | 0.252 | 0.308 | 0.250 | 0.308 |
| ILI | 2.347 | 1.089 | 2.336 | 1.086 | 2.292 | 1.069 | 1.580 | 0.852 | 1.520 | 0.825 | 1.496 | 0.807 | 2.438 | 0.955 | 2.328 | 0.984 | 2.114 | 0.900 | 1.383 | 0.718 | 1.366 | 0.713 | 1.365 | 0.712 |
| Exchange | 0.369 | 0.418 | 0.362 | 0.416 | 0.313 | 0.390 | 0.394 | 0.426 | 0.390 | 0.418 | 0.371 | 0.410 | 0.415 | 0.440 | 0.399 | 0.434 | 0.383 | 0.427 | 0.364 | 0.403 | 0.363 | 0.403 | 0.358 | 0.402 |

suggesting that KUP-BI can be effective without complex backbone re-tuning. In contrast, joint tuning (lightweight co-tuning with the backbone) achieves better results for almost all backbones, implying a strong interaction between the continuation-based auxiliary stream and backbone representations, and that moderate joint optimization can further unlock the full performance potential. The plugin-only setting yields moderate average gains. This may be because the auxiliary signal is injected into the feature processing pipeline rather than used as a simple post-hoc correction, making its effect more sensitive to the backbone–horizon interaction. When a single hyperparameter configuration is shared across all prediction lengths, such interaction may not be equally well matched to every horizon, resulting in improvements in some cases but slight degradation in others.

From the perspective of different backbones, DLinear and TimesNet benefit more significantly. DLinear, with relatively limited modeling capacity, is more sensitive to external structural cues and thus gains more from continuation-style auxiliary signals, especially on challenging datasets such as ETTh2 and Exchange. TimesNet shows substantial improvements on complex and unstable datasets such as ILI, suggesting that the structural constraints introduced by KUP-BI can effectively mitigate long-horizon uncertainty and error accumulation. In comparison, PatchTST and xPatch are stronger backbones with already competitive baselines, and therefore show more moderate improvements; nevertheless, they remain stable across datasets, further confirming the robustness and compatibility of KUP-BI.

### 4.3. Retrieval-Based vs. Prediction-Based Continuation Construction

In this subsection, we present prediction-based continuation construction (PBCC) as an alternative way to obtain an approximation of the post-target continuation. We train a DLinear predictor using the history segment as input and the post-target continuation as the supervised output, following the same training protocol and hyperparameters as the original DLinear backbone (Zeng et al., 2023a). After training, the predictor output is used as the estimated continuation for each input. For a fair comparison, we control all factors except the continuation-proxy construction. The results are reported in Table 2.

*Table 2.* Performance comparison of retrieval-based continuation construction and PBCC on the ETT datasets with a DLinear backbone. Results are averaged from all prediction lengths. The best and second-best results are highlighted in red and blue, respectively. Full results are listed in **Appendix F.2**.

| Model | DLinear | | | | | |
|---|---|---|---|---|---|---|
| | Ori | | + KUP-BI | | + PBCC | |
| Metric | MSE | MAE | MSE | MAE | MSE | MAE |
| ETTh1 | 0.445 | 0.454 | 0.439 | 0.450 | 0.438 | 0.449 |
| ETTh2 | 0.469 | 0.463 | 0.453 | 0.458 | 0.551 | 0.493 |
| ETTm1 | 0.359 | 0.381 | 0.359 | 0.380 | 0.358 | 0.381 |
| ETTm2 | 0.283 | 0.345 | 0.281 | 0.345 | 0.284 | 0.346 |

As shown in Table 2, the retrieval-based continuation construction achieves slightly better overall performance than PBCC on the ETT datasets. In particular, KUP-BI yields clear improvements over PBCC on the more challenging ETTh2 dataset, where PBCC shows a large performance drop compared with both the original backbone and KUP-BI. This suggests that directly predicting continuations can be unstable and may introduce additional errors into the auxiliary stream, especially when the future dynamics are more complex. In contrast, retrieval-based construction benefits from real continuations observed in the training data, which provides more reliable evolution patterns and leads to more stable gains. Overall, these results support our choice of using retrieval-based continuation construction.

### 4.4. Ratio vs. Residual Comparison

In this subsection, we conduct the comparison on the xPatch backbone by replacing the original ratio-based continuation construction with a residual-based continuation construction, while keeping all other settings unchanged, as shown in Table 3.

As shown in Table 3, the ratio-based continuation construction performs better than the residual-based continuation construction in most cases. One possible reason is that the residual-based representation is less scale-aware, because the same raw difference may correspond to different continuation semantics across channels or samples with different magnitudes; as a result, the constructed continuation signal can be less stable and less informative. In contrast, the ratio-style representation depends less on the raw difference itself and is therefore better suited to capture relative continuation

patterns, leading to more effective auxiliary construction in KUP-BI.

*Table 3.* Comparison of ratio-based and residual-based continuation construction on xPatch. Results are averaged over all prediction horizons. The best results are highlighted in **bold**. Full results are listed in **Appendix F.3**.

| Model | xPatch | | | |
| --- | --- | --- | --- | --- |
| | Ratio | | Residual | |
| Metric | MSE | MAE | MSE | MAE |
| ETTh1 | **0.431** | **0.438** | 0.488 | 0.462 |
| ETTh2 | **0.340** | **0.382** | 0.341 | 0.383 |
| ETTm1 | **0.352** | **0.372** | 0.352 | 0.372 |
| ETTm2 | **0.252** | **0.308** | 0.252 | 0.308 |
| ILI | **1.366** | **0.713** | 1.382 | 0.716 |
| Exchange | **0.363** | **0.403** | 0.367 | 0.405 |

## 4.5. Ablation Study

We conduct an ablation study on the xPatch backbone, including both **component-level** and **parameter-level** ablations, while keeping all other training settings and hyperparameters fixed.

**Component-level ablations** Specifically, we consider four structural variants. **Concatenation** removes gated fusion and directly appends the auxiliary input to the end of the original input sequence, testing whether an explicit fusion mechanism is necessary. **Random Retrieval** replaces correlation-based retrieval with random sampling, which evaluates whether the gains truly come from high-quality retrieval. **Direct Continuation (DC)** changes the auxiliary construction by replacing the ratio-based post-target continuation modeling with directly using the ground-truth continuation associated with retrieved similar histories in the training set, which assesses the benefit of ratio-style evolution modeling. **Target** replaces the values used as auxiliary information from post-target continuations to the corresponding target segments, which evaluates whether continuation-style cues are more effective than target-level auxiliary signals.

*Table 4.* Component-level ablations of KUP-BI on the xPatch backbone. Results are averaged over all prediction horizons. The best and second-best results are highlighted in red and blue, respectively. Full results are listed in **Appendix F.4**.

| Model | xPatch | | | | | | | | | |
| --- | --- | --- | --- | --- | --- | --- | --- | --- | --- | --- |
| | KUP-BI | | Concatenation | | Random Retrieval | | DC | | Target | |
| Metric | MSE | MAE | MSE | MAE | MSE | MAE | MSE | MAE | MSE | MAE |
| ETTh1 | 0.431 | 0.438 | 0.411 | 0.433 | 0.443 | 0.440 | 0.462 | 0.452 | 0.466 | 0.462 |
| ETTh2 | 0.340 | 0.382 | 0.349 | 0.389 | 0.341 | 0.383 | 0.341 | 0.383 | 0.340 | 0.383 |
| ETTm1 | 0.352 | 0.372 | 0.388 | 0.400 | 0.352 | 0.372 | 0.352 | 0.372 | 0.352 | 0.372 |
| ETTm2 | 0.252 | 0.308 | 0.288 | 0.338 | 0.252 | 0.308 | 0.252 | 0.308 | 0.253 | 0.308 |
| ILI | 1.366 | 0.713 | 1.713 | 0.827 | 1.378 | 0.715 | 1.393 | 0.720 | 1.382 | 0.718 |
| Exchange | 0.363 | 0.403 | 0.376 | 0.407 | 0.366 | 0.404 | 0.368 | 0.405 | 0.355 | 0.398 |

As shown in Table 4, KUP-BI generally outperforms the ablated variants across most datasets and metrics. First, **Concatenation** underperforms KUP-BI on most datasets, except for ETTh1, suggesting that simply appending aux-

iliary inputs is not a consistently effective way to incorporate auxiliary information. This supports the benefit of gated fusion in controlling the contributions of the main and auxiliary streams. Second, **Random Retrieval** performs worse than or at best comparably to KUP-BI on most datasets, suggesting that the benefit does not come from merely adding extra inputs, but from retrieving structurally relevant candidates. Third, the **Target** variant is competitive in several settings and even outperforms KUP-BI on Exchange, but it is weaker on ETTh1 and ILI, suggesting that target-level auxiliary information is less consistently effective than continuation-based information in our framework. Finally, **Direct Continuation** can also be competitive in a few cases but is overall less stable, implying that directly reusing raw continuations is more sensitive to misalignment. In contrast, KUP-BI models relative continuation patterns through ratio-style construction and further applies clipping and alignment, leading to more consistent and robust improvements.

**Parameter-level ablations** Specifically, we consider three ablation variants at the parameter level. **w/o $\alpha$** removes the residual weight $\alpha$, so the model no longer explicitly enforces the constraint that the backbone stream should dominate the fusion. **w/o $\tau$** sets $\tau$ to 1, which disables temperature scaling and makes the candidate weighting closer to a standard softmax. **w/o Top-$k$** keeps only the single most similar candidate (Top-$k = 1$), which tests whether aggregating multiple retrieved candidates improves stability and generalization. With these controlled variants, we aim to clarify the role of $\alpha$, $\tau$, and Top-$k$ in the overall framework and their impact on forecasting performance

*Table 5.* Parameter-level ablations of KUP-BI on the xPatch backbone. Results are averaged from all prediction lengths. The best results are highlighted in **bold**. Full results are listed in **Appendix F.4**.

| Model | xPatch | | | | | | | |
| --- | --- | --- | --- | --- | --- | --- | --- | --- |
| | KUP-BI | | w/o $\alpha$ | | w/o $\tau$ | | w/o Top-$k$ | |
| Metric | MSE | MAE | MSE | MAE | MSE | MAE | MSE | MAE |
| ETTh1 | **0.431** | **0.438** | 0.457 | 0.463 | 0.441 | 0.440 | **0.431** | **0.438** |
| ETTh2 | **0.340** | **0.382** | 0.352 | 0.401 | **0.340** | **0.382** | 0.341 | 0.383 |
| ETTm1 | **0.352** | **0.372** | 0.412 | 0.414 | **0.352** | **0.372** | **0.352** | **0.372** |
| ETTm2 | **0.252** | **0.308** | 0.280 | 0.339 | **0.252** | **0.308** | **0.252** | **0.308** |
| ILI | **1.366** | **0.713** | 1.929 | 0.887 | 1.382 | 0.716 | 1.389 | 0.719 |
| Exchange | **0.363** | **0.403** | 0.376 | 0.415 | **0.363** | **0.403** | 0.366 | 0.404 |

As shown in Table 5, removing or weakening these design elements generally degrades the performance of KUP-BI to varying degrees, with $\alpha$ being the most critical. Removing $\alpha$ (**w/o $\alpha$**) leads to clear degradation across datasets. For example, the MSE on ETTh1 increases from 0.431 to 0.457, ETTm1 from 0.352 to 0.412, and ILI sharply from 1.366 to 1.929. This indicates that $\alpha$ provides an important residual constraint that helps control the contribution of the auxiliary stream and preserves the backbone prediction as the dominant component, thereby improving robustness. In comparison, the impact of $\tau$ is milder. Setting $\tau$ to 1 causes

only small changes on most datasets, but still leads to noticeable drops on ETTh1 and ILI, such as the increase on ILI from 1.366 to 1.382. This suggests that temperature scaling mainly affects the sharpness of candidate weighting and appears more beneficial on more challenging datasets. Finally, using Top-$k$ = 1 (**w/o Top-$k$**) results in small degradations on several datasets, such as ETTh2 from 0.340 to 0.341, ILI from 1.366 to 1.389, and Exchange from 0.363 to 0.366, while remaining neutral on others. This suggests that aggregating multiple candidates reduces the risk of relying on a single retrieved sample and yields a more stable auxiliary evolution pattern. Overall, $\alpha$ is essential for stable gains, while $\tau$ and Top-$k$ further improve the reliability of retrieval weighting and aggregation, leading to more consistent performance.

### 4.6. Hyperparameter Sensitivity

Figure 3 examines the sensitivity of KUP-BI to three key hyperparameters on ETTh1 with a DLinear backbone.

**Effect of Top-$k$.** Varying Top-$k$ among $\{1, 3, 5, 7, 9\}$ results in nearly unchanged MSE across all horizons, indicating that KUP-BI is largely insensitive to the choice of Top-$k$.

**Effect of $\alpha$.** Performance is more sensitive to $\alpha$, which controls the fusion strength between the historical stream and the auxiliary continuation stream. As $\alpha$ increases, MSE generally decreases and achieves the best performance around $\alpha \approx 0.7 \sim 0.8$, especially for longer horizons. This indicates that a balanced fusion, where the backbone prediction remains dominant while auxiliary guidance is preserved, is crucial for stable improvements.

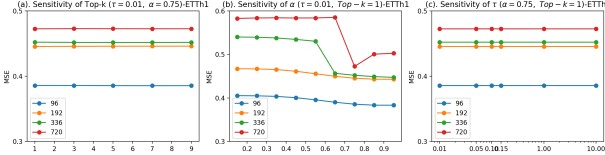

*Figure 3.* Sensitivity analysis of KUP-BI hyperparameters on ETTh1 with a DLinear backbone. We vary Top-$k$ (with fixed $\tau = 0.01$, $\alpha = 0.75$), $\alpha$ (with fixed $\tau = 0.01$, Top-$k$=1), and $\tau$ (with fixed $\alpha = 0.75$, Top-$k$=1). Results are reported in MSE under prediction lengths 96/192/336/720.

**Effect of $\tau$.** With Top-$k$ = 1, the continuation construction reduces to a single retrieved candidate, making the softmax weight invariant to $\tau$. Consequently, varying $\tau$ does not affect the auxiliary sequence and leads to nearly identical MSE curves.

Overall, $\alpha$ is the primary factor influencing performance, supporting a default choice of $\alpha = 0.75$, Top-$k$=1, and $\tau = 0.01$ on ETTh1 with DLinear. More sensitivity analysis results are provided in Appendix G.

### 4.7. Forecasting Visualization

From Figure 4, we observe that the ground truth exhibits many sharp spikes and irregular jumps in the forecasting horizon, making the overall sequence highly complex. In contrast, DLinear produces smoother predictions and shows clear long-horizon underestimation and mean drift. After introducing KUP-BI, the predicted curve matches the central level of the ground truth better in the long horizon, and its amplitude and rhythm of periodic fluctuations are also closer to the ground truth. These results indicate that the continuation-style auxiliary information effectively alleviates error accumulation and trend bias in long-horizon forecasting. It is also worth noting that both methods still struggle to accurately capture sudden spikes, suggesting that such extreme fluctuations remain a challenge for long-term forecasting.

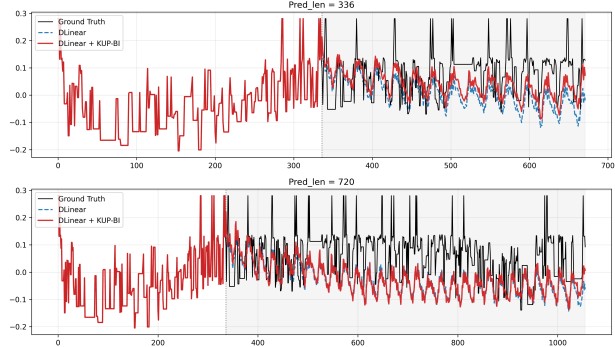

*Figure 4.* Prediction curves of DLinear and DLinear with KUP-BI under different prediction lengths.

## 5. Conclusion and Future Work

We presented KUP-BI, a knowledge-utilization paradigm that augments time series forecasting with a continuation-style auxiliary stream. The auxiliary stream is constructed from training-only chains using simple ratio-style transformations, and is fused with the main stream through a lightweight feature-level gated module. Across multiple datasets and backbones, KUP-BI achieves small but consistent error reductions with modest computational overhead, suggesting that leveraging post-target continuations from the training data provides a useful structural bias for long-horizon forecasting.

KUP-BI also has limitations. First, the current retrieval strategy is relatively simple and does not explicitly handle phase shifts, which may lead to imperfect matches and noisy auxiliary signals. Although our design alleviates this issue to some extent by distilling relative evolution cues and using feature-level gating to suppress unreliable proxies (see Appendix E for an analysis of the relationship between retrieval quality and model gains), explicitly addressing phase shifts remains necessary to achieve better prediction accuracy. Second, to fully unlock its potential, KUP-BI may

benefit from backbone-specific tuning rather than being a purely plug-and-play method. This can increase the training cost and may limit its direct applicability to large foundation models. Addressing these limitations will be an important direction for future work.

## Acknowledgements

L. Chong, Y. Zhou, and H. Li were supported in part by the National Natural Science Foundation of China (NSFC) under Grant No. 62171302 and by the Sichuan Science and Technology Program under Grant No. 2023NSFSC1965. Pengyang Wang was supported in part by the Science and Technology Development Fund, Macao SAR, under Grant Nos. 001/2024/SKL, 0002/2025/EQP, and 0072/2025/AMJ, and by the University of Macau under Grant No. MYRG-GRG2025-00241-IOTSC.

## Impact Statement

This paper aims to advance the field of long-horizon time series forecasting, which can provide decision support for domains such as energy management, transportation planning, environmental monitoring, and resource allocation. However, inaccurate forecasts may lead to undesirable decisions, especially in high-stakes real-world applications. Our work seeks to improve the reliability of long-term forecasting by leveraging post-target continuation to better model long-range temporal dependencies. We do not anticipate that this work will raise significant ethical concerns beyond the general risks associated with automated forecasting systems.

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

# Appendices

## A. Dataset Descriptions

In this paper, we consider six datasets to verify the effectiveness of the method proposed in this paper. Their details are as follows.

**ETT (Electricity Transformer Temperature)** (Zhou et al., 2021): The ETT dataset consists of two subsets with an hourly granularity ("h") and two subsets with a 15-minute granularity ("m"). These datasets contain electricity transformer data collected from two different counties between July 2016 and July 2018, with seven recorded features. The suffixes "1" and "2" distinguish the two regions where the data were collected.

**Exchange** (Lai et al., 2018): It contains the daily exchange rates of eight countries from 1990 to 2016, including Australia, the United Kingdom, Canada, Switzerland, China, Japan, New Zealand and Singapore.

**ILI** [2]: The ILI dataset reports the proportion of patients diagnosed with influenza-like illness relative to the total number of patients. It provides weekly records collected by the U.S. Centers for Disease Control and Prevention (CDC) spanning the years 2002 to 2021.

The data processing and train–validation–test splitting schemes described in earlier literature (Nie et al., 2023; Zeng et al., 2023a; Wu et al., 2023) are followed in this paper. For the ETT dataset, the split ratio is 6:2:2, while for the other datasets it is 7:1:2. Details of the datasets are provided in Table 6.

*Table 6.* Detailed dataset descriptions.

| Dataset | Information | Datset Size | Length | Channel | Frequency |
|---------|-------------|-------------|--------|---------|-----------|
| ETTh1 | | (8545, 2881, 2881) | 17420 | | 1 hour |
| ETTh2 | Energy | | | 7 | |
| ETTm1 | | (34465, 11521, 11521) | 69680 | | 15 min |
| ETTm2 | | | | | |
| ILI | Healthcare | (617, 74, 170) | 966 | 7 | 1 Week |
| Exchange | Finance | (5120, 665, 1422) | 7588 | 8 | 1 day |

## B. Backbone Models

In this paper, we select one representative state-of-the-art backbone from each of the three classic architecture families: the Transformer-based PatchTST (Nie et al., 2023), the MLP-based DLinear (Zeng et al., 2023a), and the CNN-based TimesNet (Wu et al., 2023). We additionally include a hybrid dual-stream (MLP+CNN) non-Transformer model, xPatch (Stitsyuk & Choi, 2025). All backbones are instantiated from their official codebases, and we run our experiments using the author-released hyperparameter configurations.

**PatchTST**[3]: A model based on patch learning that adopts a channel-independent approach for multivariate time-series modeling.

**DLinear**[4]: A streamlined model whose learning process involves only decomposition and linear layers.

**TimesNet**[5]: It converts one-dimensional time series into two-dimensional representations to model multi-periodic patterns and leverages CNNs to capture dependencies both across and within periods.

**xPatch** [6]: A dual-stream CNN–MLP architecture that applies exponential seasonal-trend decomposition, patching, and channel-independence to capture both linear and nonlinear patterns in multivariate time-series forecasting.

---

[2]https://gis.cdc.gov/grasp/fluview/fluportaldashboard.html
[3]https://github.com/yuqinie98/PatchTST
[4]https://github.com/cure-lab/LTSF-Linear
[5]https://github.com/thuml/Time-Series-Library
[6]https://github.com/stitsyuk/xPatch/tree/main

## C. Implementation Details

Considering that the results of TimesNet (Wu et al., 2023) and xPatch (Stitsyuk & Choi, 2025) exhibit slight run-to-run variance, all reported numbers in this paper are averaged over three independent runs. In particular, for xPatch, we follow the authors'recommended configuration in the script *xPatch_fair* (among the three scripts they provide).

**Fusion point** We keep each backbone architecturally unchanged and only attach a lightweight dual-stream fusion block at a single feature interface where the historical stream is already formed. Specifically: (i) DLinear (Zeng et al., 2023a) and xPatch (Stitsyuk & Choi, 2025) both adopt a seasonal–trend two-stream decomposition, and we fuse the historical stream with the continuation-style auxiliary stream after the decomposition stage; (ii) PatchTST (Nie et al., 2023) fuses the historical stream and the auxiliary stream after patch projection and before the Transformer encoder; (iii) TimesNet (Wu et al., 2023) fuses them after the DataEmbedding layer and before the TimesBlocks. The fusion gate is lightweight and broadcastable, while its exact instantiation follows the corresponding backbone representation. The same fusion rule is used across all backbones.

In our experiments, the static gate is used by default. Dynamic gating is enabled only for DLinear on ILI, PatchTST on ILI, and xPatch on ETTh2; all other dataset–backbone combinations use the default static-gate setting.

**Hyperparameter Tuning Protocols.** In our experiments, we use the official implementations released by the authors of each backbone method. For fair comparison, when augmenting a backbone forecaster with KUP-BI, we consider two tuning protocols: **Plugin-only tuning** and **Light joint tuning**.

**(1) Plugin-only tuning.** We keep the backbone hyperparameters fixed to their official settings and fix the fusion coefficient to $\alpha = 0.9$. We then perform a two-stage grid search over KUP-BI-specific hyperparameters: (i) we grid-search Top-$k \in \{1, 3, 5, 7, 9\}$ and $\tau \in \{0.01, 0.05, 0.1, 0.15, 1.0, 5.0, 10.0\}$ to obtain the best Top-$k^*$ and $\tau^*$; (ii) with Top-$k^*$ and $\tau^*$ fixed, we grid-search $\alpha \in \{0.15, 0.25, 0.35, 0.45, 0.55, 0.65, 0.75, 0.85, 0.9, 0.95, 1.0\}$ to determine the final $\alpha$. This protocol provides a controlled comparison by isolating the effect of KUP-BI without altering the backbone configuration.

*Table 7.* Average inference time per batch (ms/batch) for DLinear and DLinear with KUP-BI. The best results are highlighted in **bold**.

| Dataset | SeqLen | PredLen | BatchSize | DLinear | Dlinear+KUP-BI |
|---------|--------|---------|-----------|---------|----------------|
| ETTh1 | 336 | 96 | 32 | **0.373** | 0.611 |
| | 336 | 192 | 32 | **0.404** | 0.614 |
| | 336 | 336 | 32 | **0.362** | 0.597 |
| | 336 | 720 | 32 | **0.404** | 0.605 |
| ETTh2 | 336 | 96 | 32 | **0.360** | 0.619 |
| | 336 | 192 | 32 | **0.428** | 0.612 |
| | 336 | 336 | 32 | **0.356** | 0.592 |
| | 336 | 720 | 32 | **0.386** | 0.624 |
| ETTm1 | 336 | 96 | 8 | **0.300** | 0.599 |
| | 336 | 192 | 8 | **0.302** | 0.603 |
| | 336 | 336 | 8 | **0.291** | 0.581 |
| | 336 | 720 | 8 | **0.346** | 0.593 |
| ETTm2 | 336 | 96 | 32 | **0.344** | 0.627 |
| | 336 | 192 | 32 | **0.360** | 0.617 |
| | 336 | 336 | 32 | **0.329** | 0.589 |
| | 336 | 720 | 32 | **0.337** | 0.599 |
| ILI | 104 | 24 | 32 | **0.432** | 0.896 |
| | 104 | 36 | 32 | **0.442** | 0.958 |
| | 104 | 48 | 32 | **0.435** | 0.939 |
| | 104 | 60 | 32 | **0.437** | 0.946 |
| Exchange | 336 | 96 | 8 | **0.307** | 0.590 |
| | 336 | 192 | 8 | **0.309** | 0.589 |
| | 336 | 336 | 32 | **0.399** | 0.581 |
| | 336 | 720 | 32 | **0.471** | 0.575 |

**(2) Light joint tuning.** Under this protocol, while keeping the backbone architecture unchanged, we tune a small set of the backbone training hyperparameters (*e.g.*, learning rate and batch size) within a limited range. Meanwhile, we search KUP-BI hyperparameters more flexibly to reflect the method's achievable upper bound under reasonable tuning. Specifically, compared to Plugin-only tuning, we only expand the search range of Top-$k$ to the integer interval $[1, 10]$. The ranges of the other two KUP-BI hyperparameters remain the same as in Plugin-only tuning, *i.e.*, $\tau \in \{0.01, 0.05, 0.1, 0.15, 1.0, 5.0, 10.0\}$ and $\alpha \in \{0.15, 0.25, 0.35, 0.45, 0.55, 0.65, 0.75, 0.85, 0.9, 0.95\}$. We use Optuna with a TPE sampler for automated search (50 trials in total, including 30 startup trials with random sampling) (Akiba et al., 2019); in each trial, we evaluate the sampled configuration across multiple prediction horizons (pred_len) and minimize the average error over these horizons.

## D. Inference Efficiency and Runtime Overhead

Table 7 reports the average inference batch time (ms/batch) measured on the test set for DLinear and DLinear with KUP-BI across multiple datasets and prediction lengths. Overall, adding KUP-BI increases the batch time compared with the original DLinear, since it introduces auxiliary construction based on retrieval and additional fusion operations. However, the overhead is stable across settings and remains small, typically below 1 ms per batch in our experiments, indicating that KUP-BI is computationally lightweight and practical to use as a plugin. Moreover, the batch time does not increase sharply with longer prediction lengths, suggesting that the extra cost is mainly introduced by the auxiliary processing.

## E. Retrieval Quality and Forecasting Gains across Horizons

Table 8 presents the retrieved quality of KUP-BI on DLinear and its impact on forecasting performance across different prediction lengths. Overall, the benefit of KUP-BI does not increase monotonically with the prediction length; instead, it exhibits clear dataset- and horizon-dependent behavior. On some datasets, the gains mainly emerge in more challenging long-horizon settings. For example, on ETTh2, DLinear+KUP-BI achieves consistent improvements at prediction lengths 192, 336, and 720, with the largest gain at 336, where the MSE and MAE are reduced by 6.263% and 2.748%, respectively. On ETTh1, by contrast, the gains are limited at short and medium horizons and even show slight degradation in some cases, while a clear improvement is observed only at the longest horizon 720, with 6.349% and 5.049% reductions in MSE and MAE, respectively. Similarly, for ILI and Exchange, the gains are mainly observed at the longest prediction lengths, whereas the improvements at shorter horizons are marginal or even slightly negative. These results suggest that when the forecasting task becomes more challenging and the backbone is more prone to error accumulation, the auxiliary continuation provided by KUP-BI is more likely to be helpful and yield more substantial performance gains.

More importantly, within the same data family, retrieved quality and forecasting gain exhibit a consistent correspondence. Taking ETTh as an example, ETTh2 has clearly better retrieved quality than ETTh1, as reflected by lower average retrieved MSE/MAE and higher Corr, and it also achieves larger and more stable improvements overall. In contrast, ETTh1, whose retrieved quality is weaker, shows a clear gain only at the longest horizon. A similar pattern is also observed for ETTm: ETTm2 has better retrieved quality than ETTm1, and correspondingly shows more positive error reductions across multiple horizons, whereas ETTm1 remains almost unchanged. This observation indicates that, within the same type of time-series data, higher-quality retrieval results are more likely to translate into larger forecasting gains; that is, a more accurate and more correlated auxiliary continuation can more effectively compensate for the backbone's limitations in modeling future segments. Overall, Table 8 suggests that the effectiveness of KUP-BI is closely related to both the forecasting scenario and the quality of the retrieved inputs, with better retrieved quality within the same data family generally leading to larger and more stable forecasting gains.

## F. Full Results

### F.1. Full results of the long-term forecasting task

Table 9 presents the complete results corresponding to the main experiment in Table 1. Overall, KUP-BI improves forecasting performance across most datasets, backbones, and prediction lengths, further supporting its effectiveness.

### F.2. Full Results for Retrieval-Based vs. Prediction-Based Continuation Construction

Table 10 reports the full comparison between retrieval-based continuation construction and prediction-based continuation construction (PBCC) on the ETT datasets using DLinear as the backbone. Overall, the retrieval-based method in KUP-BI achieves slightly better average performance.

### F.3. Full Results for Ratio vs. Residual Comparison

Table 11 reports the full comparison between the ratio-based continuation construction and the residual-based continuation construction on the xPatch backbone across different forecasting horizons. All other settings are kept unchanged. Overall, the ratio-based version achieves better or comparable performance in most cases, which further supports our choice of using ratio-style continuation construction in KUP-BI.

*Table 8.* Retrieved quality and forecasting benefits across horizons on DLinear. We report the relative error reduction (%) of DLinear+KUP-BI over DLinear across different forecasting horizons. Retrieved quality is measured by MSE, MAE, and Pearson correlation coefficient (Corr ) between the estimated auxiliary input (*i.e.*, the estimated post-target continuation) and the ground-truth post-target continuation over the corresponding future segment. Positive values in the improvement columns indicate lower forecasting error compared with the original backbone.

| Dataset | Pred length | DLinear | | DLinear+KUP-BI | | Improvement (%) | | Retrieved quality | | |
|---|---|---|---|---|---|---|---|---|---|---|
| | | MSE | MAE | MSE | MAE | MSE | MAE | MSE | MAE | Corr |
| ETTh1 | 96 | 0.384 | 0.405 | 0.386 | 0.407 | -0.521% | -0.494% | 0.970 | 0.649 | 0.559 |
| | 192 | 0.443 | 0.450 | 0.445 | 0.451 | -0.451% | -0.222% | 0.966 | 0.653 | 0.556 |
| | 336 | 0.447 | 0.448 | 0.452 | 0.452 | -1.119% | -0.893% | 0.984 | 0.670 | 0.555 |
| | 720 | 0.504 | 0.515 | 0.472 | 0.489 | 6.349% | 5.049% | 1.101 | 0.721 | 0.504 |
| | Avg | | | | | 1.065% | 0.860% | | | |
| ETTh2 | 96 | 0.290 | 0.353 | 0.294 | 0.357 | -1.379% | -1.133% | 0.597 | 0.528 | 0.825 |
| | 192 | 0.388 | 0.422 | 0.377 | 0.418 | 2.835% | 0.948% | 0.619 | 0.545 | 0.820 |
| | 336 | 0.463 | 0.473 | 0.434 | 0.460 | 6.263% | 2.748% | 0.695 | 0.581 | 0.802 |
| | 720 | 0.733 | 0.606 | 0.707 | 0.597 | 3.547% | 1.485% | 0.763 | 0.635 | 0.771 |
| | Avg | | | | | 2.817% | 1.012% | | | |
| ETTm1 | 96 | 0.301 | 0.345 | 0.301 | 0.345 | 0.000% | 0.000% | 1.077 | 0.651 | 0.532 |
| | 192 | 0.336 | 0.366 | 0.335 | 0.366 | 0.298% | 0.000% | 1.037 | 0.649 | 0.550 |
| | 336 | 0.372 | 0.389 | 0.372 | 0.389 | 0.000% | 0.000% | 0.864 | 0.604 | 0.634 |
| | 720 | 0.427 | 0.423 | 0.427 | 0.423 | 0.000% | 0.000% | 0.949 | 0.636 | 0.600 |
| | Avg | | | | | 0.074% | 0.000% | | | |
| ETTm2 | 96 | 0.172 | 0.267 | 0.172 | 0.268 | 0.000% | -0.375% | 0.566 | 0.495 | 0.843 |
| | 192 | 0.237 | 0.314 | 0.233 | 0.315 | 1.688% | -0.318% | 0.607 | 0.514 | 0.837 |
| | 336 | 0.295 | 0.359 | 0.294 | 0.359 | 0.339% | 0.000% | 0.623 | 0.518 | 0.841 |
| | 720 | 0.427 | 0.439 | 0.424 | 0.438 | 0.703% | 0.228% | 0.689 | 0.565 | 0.823 |
| | Avg | | | | | 0.682% | -0.116% | | | |
| ILI | 24 | 2.280 | 1.061 | 2.316 | 1.076 | -1.579% | -1.414% | 6.687 | 1.901 | 0.159 |
| | 36 | 2.235 | 1.059 | 2.263 | 1.062 | -1.253% | -0.283% | 7.387 | 1.963 | 0.064 |
| | 48 | 2.298 | 1.079 | 2.327 | 1.088 | -1.262% | -0.834% | 6.675 | 1.832 | 0.232 |
| | 60 | 2.573 | 1.157 | 2.439 | 1.118 | 5.208% | 3.371% | 7.132 | 1.866 | 0.120 |
| | Avg | | | | | 0.279% | 0.210% | | | |
| Exchange | 96 | 0.085 | 0.209 | 0.085 | 0.209 | 0.000% | 0.000% | 0.899 | 0.736 | 0.833 |
| | 192 | 0.162 | 0.296 | 0.162 | 0.296 | 0.000% | 0.000% | 1.154 | 0.855 | 0.807 |
| | 336 | 0.333 | 0.441 | 0.343 | 0.451 | -3.003% | -2.268% | 1.617 | 1.016 | 0.757 |
| | 720 | 0.898 | 0.725 | 0.858 | 0.707 | 4.454% | 2.483% | 3.197 | 1.428 | 0.596 |
| | Avg | | | | | 0.363% | 0.054% | | | |

## F.4. Full Ablation Results

Tables 12 and 13 report the full results of the **component-level ablations** and the **parameter-level ablations** of KUP-BI on the xPatch backbone, respectively.

For the **component-level ablations**, the original KUP-BI yields the most consistent performance across most datasets and prediction lengths, although some variants are competitive or even better in a few settings. In general, **Concatenation** and **Random Retrieval** perform worse or less consistently, highlighting the importance of controlled fusion and high-quality retrieval. The **Direct Continuation** variant can be competitive in some cases, but its performance is less stable across datasets, which supports the use of ratio-based continuation construction in KUP-BI. The **Target** variant is also competitive in several settings, but is less consistent overall than the original KUP-BI, suggesting that target-based auxiliary signals are less robust than continuation-style cues in our framework.

For the **parameter-level ablations**, removing $\alpha$ leads to the clearest and most consistent degradation across datasets, indicating that the residual constraint is crucial for stable fusion. In comparison, fixing $\tau$ to 1 or reducing Top-$k$ to 1 usually causes only mild degradation or ties with the original setting, suggesting that temperature scaling and multi-candidate aggregation mainly improve the robustness of retrieval weighting.

## G. Additional Sensitivity Analysis of KUP-BI Hyperparameters

Figures 5 and 6 present the sensitivity analysis of KUP-BI on the DLinear and xPatch backbones under different prediction lengths. Overall, the performance is insensitive to Top-$k$, as varying Top-$k$ leads to nearly unchanged MSE across horizons, indicating that the retrieval stage is robust once a reasonable candidate set is available. In contrast, $\alpha$ has a clear impact, and moderate values usually yield better performance, showing that the fusion strength between the backbone stream and the auxiliary continuation stream is an important factor. Finally, the results are relatively stable with respect to $\tau$ in most settings, suggesting that temperature scaling mainly provides mild adjustments rather than dominating the performance.

*Table 9.* Full results of the long-term forecasting task. "Avg" means the average results from all four prediction lengths. The best and second-best results are highlighted in red and blue, respectively.

| Model | | DLinear Ori | | + KUP-BI (Plugin-only) | | + KUP-BI (Joint-tune) | | PatchTST Ori | | + KUP-BI (Plugin-only) | | + KUP-BI (Joint-tune) | | TimesNet Ori | | + KUP-BI (Plugin-only) | | + KUP-BI (Joint-tune) | | xPatch Ori | | + KUP-BI (Plugin-only) | | + KUP-BI (Joint-tune) | |
|---|---|---|---|---|---|---|---|---|---|---|---|---|---|---|---|---|---|---|---|---|---|---|---|---|---|
| Metric | | MSE | MAE | MSE | MAE | MSE | MAE | MSE | MAE | MSE | MAE | MSE | MAE | MSE | MAE | MSE | MAE | MSE | MAE | MSE | MAE | MSE | MAE | MSE | MAE |
| ETTh1 96 | | 0.384 | 0.405 | 0.386 | 0.407 | 0.372 | 0.394 | 0.382 | 0.405 | 0.374 | 0.399 | 0.364 | 0.391 | 0.396 | 0.417 | 0.401 | 0.418 | 0.390 | 0.414 | 0.369 | 0.397 | 0.383 | 0.409 | 0.360 | 0.389 |
| ETTh1 192 | | 0.443 | 0.450 | 0.445 | 0.451 | 0.406 | 0.415 | 0.414 | 0.421 | 0.411 | 0.420 | 0.404 | 0.415 | 0.466 | 0.459 | 0.459 | 0.454 | 0.441 | 0.443 | 0.424 | 0.426 | 0.427 | 0.433 | 0.405 | 0.414 |
| ETTh1 336 | | 0.447 | 0.448 | 0.452 | 0.452 | 0.443 | 0.443 | 0.431 | 0.436 | 0.425 | 0.430 | 0.434 | 0.440 | 0.506 | 0.477 | 0.486 | 0.467 | 0.507 | 0.480 | 0.450 | 0.437 | 0.450 | 0.444 | 0.423 | 0.428 |
| ETTh1 720 | | 0.504 | 0.515 | 0.472 | 0.489 | 0.479 | 0.495 | 0.449 | 0.466 | 0.443 | 0.461 | 0.432 | 0.456 | 0.521 | 0.497 | 0.507 | 0.492 | 0.493 | 0.484 | 0.533 | 0.494 | 0.462 | 0.465 | 0.449 | 0.458 |
| ETTh1 Avg | | 0.445 | 0.454 | 0.439 | 0.450 | 0.425 | 0.437 | 0.419 | 0.432 | 0.413 | 0.428 | 0.409 | 0.425 | 0.472 | 0.463 | 0.463 | 0.458 | 0.458 | 0.455 | 0.444 | 0.438 | 0.431 | 0.438 | 0.409 | 0.422 |
| ETTh2 96 | | 0.290 | 0.353 | 0.294 | 0.357 | 0.283 | 0.347 | 0.275 | 0.337 | 0.275 | 0.337 | 0.272 | 0.334 | 0.336 | 0.372 | 0.312 | 0.357 | 0.308 | 0.358 | 0.275 | 0.333 | 0.272 | 0.332 | 0.275 | 0.334 |
| ETTh2 192 | | 0.388 | 0.422 | 0.377 | 0.418 | 0.345 | 0.393 | 0.338 | 0.378 | 0.337 | 0.377 | 0.334 | 0.377 | 0.423 | 0.426 | 0.404 | 0.409 | 0.415 | 0.418 | 0.337 | 0.375 | 0.342 | 0.377 | 0.338 | 0.376 |
| ETTh2 336 | | 0.463 | 0.473 | 0.434 | 0.460 | 0.400 | 0.434 | 0.329 | 0.379 | 0.325 | 0.382 | 0.324 | 0.382 | 0.444 | 0.444 | 0.439 | 0.447 | 0.452 | 0.449 | 0.365 | 0.398 | 0.359 | 0.396 | 0.360 | 0.397 |
| ETTh2 720 | | 0.733 | 0.606 | 0.707 | 0.597 | 0.550 | 0.528 | 0.379 | 0.422 | 0.380 | 0.422 | 0.375 | 0.419 | 0.457 | 0.464 | 0.455 | 0.462 | 0.428 | 0.448 | 0.391 | 0.426 | 0.387 | 0.424 | 0.381 | 0.420 |
| ETTh2 Avg | | 0.469 | 0.463 | 0.453 | 0.458 | 0.394 | 0.426 | 0.330 | 0.379 | 0.329 | 0.379 | 0.326 | 0.378 | 0.415 | 0.426 | 0.402 | 0.419 | 0.401 | 0.418 | 0.342 | 0.383 | 0.340 | 0.382 | 0.339 | 0.382 |
| ETTm1 96 | | 0.301 | 0.345 | 0.301 | 0.345 | 0.300 | 0.344 | 0.292 | 0.343 | 0.294 | 0.345 | 0.295 | 0.345 | 0.338 | 0.375 | 0.337 | 0.376 | 0.332 | 0.375 | 0.289 | 0.332 | 0.289 | 0.332 | 0.288 | 0.333 |
| ETTm1 192 | | 0.336 | 0.366 | 0.335 | 0.366 | 0.335 | 0.366 | 0.336 | 0.371 | 0.333 | 0.372 | 0.329 | 0.368 | 0.403 | 0.408 | 0.385 | 0.400 | 0.378 | 0.396 | 0.330 | 0.357 | 0.330 | 0.357 | 0.331 | 0.358 |
| ETTm1 336 | | 0.372 | 0.389 | 0.372 | 0.389 | 0.373 | 0.390 | 0.366 | 0.392 | 0.364 | 0.392 | 0.362 | 0.388 | 0.421 | 0.424 | 0.423 | 0.426 | 0.410 | 0.419 | 0.364 | 0.381 | 0.364 | 0.381 | 0.363 | 0.382 |
| ETTm1 720 | | 0.427 | 0.423 | 0.427 | 0.423 | 0.425 | 0.420 | 0.418 | 0.424 | 0.415 | 0.420 | 0.413 | 0.417 | 0.498 | 0.464 | 0.483 | 0.458 | 0.500 | 0.467 | 0.426 | 0.417 | 0.426 | 0.417 | 0.418 | 0.414 |
| ETTm1 Avg | | 0.359 | 0.381 | 0.359 | 0.380 | 0.358 | 0.380 | 0.353 | 0.382 | 0.352 | 0.382 | 0.350 | 0.379 | 0.415 | 0.418 | 0.407 | 0.415 | 0.405 | 0.414 | 0.352 | 0.372 | 0.352 | 0.372 | 0.350 | 0.372 |
| ETTm2 96 | | 0.172 | 0.267 | 0.172 | 0.268 | 0.166 | 0.259 | 0.165 | 0.255 | 0.165 | 0.255 | 0.165 | 0.255 | 0.188 | 0.266 | 0.184 | 0.266 | 0.181 | 0.263 | 0.159 | 0.245 | 0.159 | 0.244 | 0.160 | 0.246 |
| ETTm2 192 | | 0.237 | 0.314 | 0.233 | 0.315 | 0.222 | 0.300 | 0.220 | 0.292 | 0.223 | 0.294 | 0.220 | 0.292 | 0.257 | 0.312 | 0.252 | 0.309 | 0.246 | 0.305 | 0.217 | 0.286 | 0.217 | 0.286 | 0.217 | 0.288 |
| ETTm2 336 | | 0.295 | 0.359 | 0.294 | 0.359 | 0.277 | 0.364 | 0.278 | 0.329 | 0.277 | 0.329 | 0.274 | 0.327 | 0.322 | 0.350 | 0.309 | 0.344 | 0.309 | 0.344 | 0.275 | 0.325 | 0.275 | 0.326 | 0.275 | 0.325 |
| ETTm2 720 | | 0.427 | 0.439 | 0.424 | 0.438 | 0.377 | 0.400 | 0.368 | 0.385 | 0.363 | 0.382 | 0.362 | 0.381 | 0.419 | 0.405 | 0.424 | 0.409 | 0.415 | 0.404 | 0.357 | 0.377 | 0.357 | 0.377 | 0.348 | 0.374 |
| ETTm2 Avg | | 0.283 | 0.345 | 0.281 | 0.345 | 0.266 | 0.330 | 0.258 | 0.315 | 0.257 | 0.315 | 0.255 | 0.314 | 0.296 | 0.333 | 0.292 | 0.332 | 0.288 | 0.329 | 0.252 | 0.308 | 0.252 | 0.308 | 0.250 | 0.308 |
| ILI 24 | | 2.280 | 1.061 | 2.316 | 1.076 | 2.224 | 1.036 | 1.584 | 0.840 | 1.637 | 0.844 | 1.409 | 0.781 | 2.662 | 0.974 | 2.629 | 1.026 | 1.840 | 0.866 | 1.334 | 0.699 | 1.325 | 0.696 | 1.323 | 0.693 |
| ILI 36 | | 2.235 | 1.059 | 2.263 | 1.062 | 2.225 | 1.057 | 1.442 | 0.831 | 1.221 | 0.743 | 1.341 | 0.765 | 2.756 | 1.010 | 2.152 | 0.950 | 2.549 | 0.973 | 1.329 | 0.683 | 1.300 | 0.675 | 1.300 | 0.675 |
| ILI 48 | | 2.298 | 1.079 | 2.327 | 1.088 | 2.266 | 1.060 | 1.685 | 0.853 | 1.578 | 0.849 | 1.596 | 0.824 | 2.299 | 0.922 | 2.332 | 0.982 | 2.145 | 0.878 | 1.358 | 0.706 | 1.368 | 0.710 | 1.373 | 0.711 |
| ILI 60 | | 2.573 | 1.157 | 2.439 | 1.118 | 2.453 | 1.121 | 1.608 | 0.886 | 1.643 | 0.862 | 1.636 | 0.860 | 2.035 | 0.912 | 2.198 | 0.977 | 1.920 | 0.881 | 1.512 | 0.785 | 1.472 | 0.773 | 1.463 | 0.770 |
| ILI Avg | | 2.347 | 1.089 | 2.336 | 1.086 | 2.292 | 1.069 | 1.580 | 0.852 | 1.520 | 0.825 | 1.496 | 0.807 | 2.438 | 0.955 | 2.328 | 0.984 | 2.114 | 0.900 | 1.383 | 0.718 | 1.366 | 0.713 | 1.365 | 0.712 |
| Exchange 96 | | 0.085 | 0.209 | 0.085 | 0.209 | 0.086 | 0.207 | 0.095 | 0.215 | 0.089 | 0.208 | 0.088 | 0.211 | 0.108 | 0.236 | 0.108 | 0.238 | 0.100 | 0.227 | 0.081 | 0.197 | 0.082 | 0.198 | 0.085 | 0.202 |
| Exchange 192 | | 0.162 | 0.296 | 0.162 | 0.296 | 0.159 | 0.289 | 0.213 | 0.330 | 0.190 | 0.310 | 0.192 | 0.315 | 0.213 | 0.335 | 0.197 | 0.322 | 0.210 | 0.333 | 0.175 | 0.296 | 0.175 | 0.296 | 0.181 | 0.302 |
| Exchange 336 | | 0.333 | 0.441 | 0.343 | 0.451 | 0.357 | 0.439 | 0.395 | 0.457 | 0.343 | 0.424 | 0.332 | 0.417 | 0.367 | 0.439 | 0.372 | 0.444 | 0.361 | 0.439 | 0.343 | 0.421 | 0.341 | 0.420 | 0.343 | 0.422 |
| Exchange 720 | | 0.898 | 0.725 | 0.858 | 0.707 | 0.652 | 0.626 | 0.872 | 0.703 | 0.936 | 0.729 | 0.872 | 0.699 | 0.971 | 0.751 | 0.919 | 0.731 | 0.860 | 0.708 | 0.857 | 0.698 | 0.854 | 0.696 | 0.822 | 0.682 |
| Exchange Avg | | 0.369 | 0.418 | 0.362 | 0.416 | 0.313 | 0.390 | 0.394 | 0.426 | 0.390 | 0.418 | 0.371 | 0.410 | 0.415 | 0.440 | 0.399 | 0.434 | 0.383 | 0.427 | 0.364 | 0.403 | 0.363 | 0.403 | 0.358 | 0.402 |

*Table 10.* Full performance comparison between retrieval-based continuation construction and prediction-based continuation construction (PBCC) on the ETT datasets using DLinear as the backbone. "Avg" means the average results from all four prediction lengths. The best and second-best results are highlighted in red and blue, respectively.

| Model | | DLinear Ori | | + KUP-BI | | + PBCC | |
|---|---|---|---|---|---|---|---|
| Metric | | MSE | MAE | MSE | MAE | MSE | MAE |
| ETTh1 | 96 | 0.384 | 0.405 | 0.386 | 0.407 | 0.385 | 0.407 |
| | 192 | 0.443 | 0.450 | 0.445 | 0.451 | 0.445 | 0.452 |
| | 336 | 0.447 | 0.448 | 0.452 | 0.452 | 0.450 | 0.451 |
| | 720 | 0.504 | 0.515 | 0.472 | 0.489 | 0.472 | 0.487 |
| | Avg | 0.445 | 0.454 | 0.439 | 0.450 | 0.438 | 0.449 |
| ETTh2 | 96 | 0.290 | 0.353 | 0.294 | 0.357 | 0.325 | 0.370 |
| | 192 | 0.388 | 0.422 | 0.377 | 0.418 | 0.451 | 0.459 |
| | 336 | 0.463 | 0.473 | 0.434 | 0.460 | 0.539 | 0.492 |
| | 720 | 0.733 | 0.606 | 0.707 | 0.597 | 0.890 | 0.652 |
| | Avg | 0.469 | 0.463 | 0.453 | 0.458 | 0.551 | 0.493 |
| ETTm1 | 96 | 0.301 | 0.345 | 0.301 | 0.345 | 0.301 | 0.345 |
| | 192 | 0.336 | 0.366 | 0.335 | 0.366 | 0.336 | 0.366 |
| | 336 | 0.372 | 0.389 | 0.372 | 0.389 | 0.371 | 0.389 |
| | 720 | 0.427 | 0.423 | 0.427 | 0.423 | 0.426 | 0.423 |
| | Avg | 0.359 | 0.381 | 0.359 | 0.380 | 0.358 | 0.381 |
| ETTm2 | 96 | 0.172 | 0.267 | 0.172 | 0.268 | 0.172 | 0.266 |
| | 192 | 0.237 | 0.314 | 0.233 | 0.315 | 0.238 | 0.314 |
| | 336 | 0.295 | 0.359 | 0.294 | 0.359 | 0.300 | 0.364 |
| | 720 | 0.427 | 0.439 | 0.424 | 0.438 | 0.428 | 0.442 |
| | Avg | 0.283 | 0.345 | 0.281 | 0.345 | 0.284 | 0.346 |

*Table 11.* Full comparison of ratio-based and residual-based continuation construction on xPatch under different forecasting horizons. "Avg" means the average results from all four prediction lengths. The best results are highlighted in **bold**.

| Model | | xPatch | | | |
| --- | --- | --- | --- | --- | --- |
| | | Ratio | | Residual | |
| Metric | | MSE | MAE | MSE | MAE |
| ETTh1 | 96 | **0.383** | **0.409** | 0.385 | 0.406 |
| | 192 | **0.427** | **0.433** | 0.468 | 0.448 |
| | 336 | **0.450** | **0.444** | 0.460 | 0.448 |
| | 720 | **0.462** | **0.465** | 0.640 | 0.546 |
| | Avg | **0.431** | **0.438** | 0.488 | 0.462 |
| ETTh2 | 96 | **0.272** | **0.332** | 0.274 | 0.333 |
| | 192 | **0.342** | **0.377** | 0.343 | 0.378 |
| | 336 | **0.359** | **0.396** | 0.360 | 0.397 |
| | 720 | **0.387** | **0.424** | 0.387 | 0.425 |
| | Avg | **0.340** | **0.382** | 0.341 | 0.383 |
| ETTm1 | 96 | **0.289** | **0.332** | **0.289** | **0.332** |
| | 192 | **0.330** | **0.357** | **0.330** | **0.357** |
| | 336 | **0.364** | **0.381** | **0.364** | **0.381** |
| | 720 | **0.426** | **0.417** | **0.426** | **0.417** |
| | Avg | **0.352** | **0.372** | **0.352** | **0.372** |
| ETTm2 | 96 | **0.159** | **0.244** | **0.159** | 0.245 |
| | 192 | **0.217** | **0.286** | **0.217** | **0.286** |
| | 336 | **0.275** | **0.326** | **0.275** | **0.326** |
| | 720 | 0.357 | 0.377 | **0.356** | **0.376** |
| | Avg | **0.252** | **0.308** | **0.252** | **0.308** |
| ILI | 24 | **1.325** | **0.696** | 1.328 | **0.696** |
| | 36 | **1.300** | **0.675** | 1.303 | 0.676 |
| | 48 | **1.368** | 0.710 | 1.369 | **0.708** |
| | 60 | **1.472** | **0.773** | 1.530 | 0.786 |
| | Avg | **1.366** | **0.713** | 1.382 | 0.716 |
| Exchange | 96 | 0.082 | **0.198** | **0.081** | **0.198** |
| | 192 | **0.175** | **0.296** | **0.175** | **0.296** |
| | 336 | **0.341** | **0.420** | 0.343 | 0.422 |
| | 720 | **0.854** | **0.696** | 0.870 | 0.704 |
| | Avg | **0.363** | **0.403** | 0.367 | 0.405 |

*Table 12.* Component-level ablations of KUP-BI on the xPatch backbone. "Avg" means the average results from all four prediction lengths. The best and second-best results are highlighted in red and blue, respectively.

| Model | | xPatch | | | | | | | | | |
| --- | --- | --- | --- | --- | --- | --- | --- | --- | --- | --- | --- |
| | | +KUP-BI | | Concatenation | | Random Retrieval | | Direct Continuation | | Target | |
| Metric | | MSE | MAE | MSE | MAE | MSE | MAE | MSE | MAE | MSE | MAE |
| ETTh1 | 96 | 0.383 | 0.409 | 0.374 | 0.409 | 0.381 | 0.404 | 0.390 | 0.411 | 0.413 | 0.434 |
| | 192 | 0.427 | 0.433 | 0.412 | 0.430 | 0.427 | 0.430 | 0.435 | 0.437 | 0.442 | 0.449 |
| | 336 | 0.450 | 0.444 | 0.427 | 0.439 | 0.444 | 0.437 | 0.463 | 0.449 | 0.440 | 0.448 |
| | 720 | 0.462 | 0.465 | 0.431 | 0.454 | 0.521 | 0.489 | 0.560 | 0.511 | 0.568 | 0.519 |
| | Avg | 0.431 | 0.438 | 0.411 | 0.433 | 0.443 | 0.440 | 0.462 | 0.452 | 0.466 | 0.462 |
| ETTh2 | 96 | 0.272 | 0.332 | 0.282 | 0.342 | 0.274 | 0.333 | 0.274 | 0.333 | 0.274 | 0.333 |
| | 192 | 0.342 | 0.377 | 0.351 | 0.384 | 0.342 | 0.377 | 0.342 | 0.377 | 0.342 | 0.377 |
| | 336 | 0.359 | 0.396 | 0.372 | 0.406 | 0.359 | 0.396 | 0.361 | 0.397 | 0.358 | 0.396 |
| | 720 | 0.387 | 0.424 | 0.391 | 0.426 | 0.387 | 0.425 | 0.387 | 0.424 | 0.387 | 0.424 |
| | Avg | 0.340 | 0.382 | 0.349 | 0.389 | 0.341 | 0.383 | 0.341 | 0.383 | 0.340 | 0.383 |
| ETTm1 | 96 | 0.289 | 0.332 | 0.345 | 0.378 | 0.289 | 0.332 | 0.289 | 0.332 | 0.289 | 0.332 |
| | 192 | 0.330 | 0.357 | 0.368 | 0.391 | 0.330 | 0.357 | 0.330 | 0.357 | 0.330 | 0.357 |
| | 336 | 0.364 | 0.381 | 0.399 | 0.405 | 0.364 | 0.381 | 0.364 | 0.381 | 0.364 | 0.381 |
| | 720 | 0.426 | 0.417 | 0.441 | 0.427 | 0.426 | 0.417 | 0.426 | 0.417 | 0.426 | 0.417 |
| | Avg | 0.352 | 0.372 | 0.388 | 0.400 | 0.352 | 0.372 | 0.352 | 0.372 | 0.352 | 0.372 |
| ETTm2 | 96 | 0.159 | 0.244 | 0.181 | 0.272 | 0.159 | 0.245 | 0.159 | 0.245 | 0.160 | 0.245 |
| | 192 | 0.217 | 0.286 | 0.241 | 0.313 | 0.218 | 0.286 | 0.218 | 0.287 | 0.218 | 0.286 |
| | 336 | 0.275 | 0.326 | 0.318 | 0.362 | 0.275 | 0.325 | 0.276 | 0.326 | 0.275 | 0.326 |
| | 720 | 0.357 | 0.377 | 0.411 | 0.406 | 0.357 | 0.377 | 0.357 | 0.377 | 0.358 | 0.377 |
| | Avg | 0.252 | 0.308 | 0.288 | 0.338 | 0.252 | 0.308 | 0.252 | 0.308 | 0.253 | 0.308 |
| ILI | 24 | 1.325 | 0.696 | 1.618 | 0.795 | 1.334 | 0.698 | 1.368 | 0.711 | 1.370 | 0.706 |
| | 36 | 1.300 | 0.675 | 1.477 | 0.748 | 1.318 | 0.680 | 1.317 | 0.680 | 1.321 | 0.680 |
| | 48 | 1.368 | 0.710 | 1.378 | 0.748 | 1.351 | 0.703 | 1.363 | 0.707 | 1.318 | 0.699 |
| | 60 | 1.472 | 0.773 | 2.378 | 1.017 | 1.507 | 0.780 | 1.526 | 0.782 | 1.519 | 0.787 |
| | Avg | 1.366 | 0.713 | 1.713 | 0.827 | 1.378 | 0.715 | 1.393 | 0.720 | 1.382 | 0.718 |
| Exchange | 96 | 0.082 | 0.198 | 0.082 | 0.199 | 0.081 | 0.198 | 0.080 | 0.197 | 0.083 | 0.202 |
| | 192 | 0.175 | 0.296 | 0.174 | 0.296 | 0.176 | 0.297 | 0.174 | 0.295 | 0.171 | 0.294 |
| | 336 | 0.341 | 0.420 | 0.325 | 0.411 | 0.343 | 0.421 | 0.347 | 0.424 | 0.331 | 0.414 |
| | 720 | 0.854 | 0.696 | 0.924 | 0.721 | 0.864 | 0.701 | 0.872 | 0.704 | 0.834 | 0.684 |
| | Avg | 0.363 | 0.403 | 0.376 | 0.407 | 0.366 | 0.404 | 0.368 | 0.405 | 0.355 | 0.398 |

*Table 13.* Parameter-level ablations of KUP-BI on the xPatch backbone. "Avg" means the average results from all four prediction lengths. The best results are highlighted in **bold**.

| Model | | xPatch | | | | | | |
|---|---|---|---|---|---|---|---|---|
| | | +KUP-BI | | w/o α | | w/o τ | | w/o Top-k | |
| Metric | | MSE | MAE | MSE | MAE | MSE | MAE | MSE | MAE |
| | 96 | 0.383 | 0.409 | 0.422 | 0.441 | **0.372** | **0.401** | 0.383 | 0.408 |
| | 192 | 0.427 | 0.433 | 0.446 | 0.452 | **0.422** | **0.427** | 0.427 | 0.433 |
| ETTh1 | 336 | 0.450 | 0.444 | 0.481 | 0.465 | 0.471 | 0.448 | **0.449** | **0.443** |
| | 720 | **0.462** | **0.465** | 0.480 | 0.493 | 0.498 | 0.482 | 0.463 | 0.466 |
| | Avg | **0.431** | **0.438** | 0.457 | 0.463 | 0.441 | 0.440 | **0.431** | **0.438** |
| | 96 | **0.272** | **0.332** | 0.292 | 0.360 | **0.272** | **0.332** | 0.274 | 0.333 |
| | 192 | **0.342** | **0.377** | 0.356 | 0.398 | **0.342** | **0.377** | 0.343 | **0.377** |
| ETTh2 | 336 | 0.359 | **0.396** | 0.356 | 0.403 | 0.359 | **0.396** | 0.360 | **0.396** |
| | 720 | **0.387** | **0.424** | 0.405 | 0.442 | **0.387** | **0.424** | **0.387** | 0.425 |
| | Avg | **0.340** | **0.382** | 0.352 | 0.401 | **0.340** | **0.382** | 0.341 | 0.383 |
| | 96 | **0.289** | **0.332** | 0.377 | 0.403 | **0.289** | **0.332** | **0.289** | **0.332** |
| | 192 | **0.330** | **0.357** | 0.430 | 0.422 | **0.330** | **0.357** | **0.330** | **0.357** |
| ETTm1 | 336 | **0.364** | **0.381** | 0.394 | 0.402 | **0.364** | **0.381** | **0.364** | **0.381** |
| | 720 | **0.426** | **0.417** | 0.449 | 0.431 | **0.426** | **0.417** | **0.426** | **0.417** |
| | Avg | **0.352** | **0.372** | 0.412 | 0.414 | **0.352** | **0.372** | **0.352** | **0.372** |
| | 96 | **0.159** | **0.244** | 0.196 | 0.285 | **0.159** | **0.244** | **0.159** | 0.245 |
| | 192 | **0.217** | **0.286** | 0.251 | 0.324 | **0.217** | **0.286** | **0.217** | **0.286** |
| ETTm2 | 336 | **0.275** | **0.326** | 0.298 | 0.352 | **0.275** | **0.326** | 0.276 | **0.326** |
| | 720 | **0.357** | **0.377** | 0.376 | 0.396 | **0.357** | **0.377** | **0.357** | **0.377** |
| | Avg | **0.252** | **0.308** | 0.280 | 0.339 | **0.252** | **0.308** | **0.252** | **0.308** |
| | 24 | **1.325** | 0.696 | 1.561 | 0.788 | 1.328 | **0.695** | 1.327 | **0.695** |
| | 36 | 1.300 | 0.675 | 1.673 | 0.812 | 1.308 | 0.675 | **1.298** | **0.674** |
| ILI | 48 | **1.368** | **0.710** | 2.596 | 1.055 | 1.370 | **0.710** | 1.374 | **0.710** |
| | 60 | **1.472** | **0.773** | 1.887 | 0.894 | 1.522 | 0.785 | 1.555 | 0.795 |
| | Avg | **1.366** | **0.713** | 1.929 | 0.887 | 1.382 | 0.716 | 1.389 | 0.719 |
| | 96 | **0.082** | **0.198** | 0.093 | 0.216 | **0.082** | **0.198** | **0.082** | 0.199 |
| | 192 | 0.175 | **0.296** | 0.186 | 0.310 | 0.175 | **0.296** | 0.174 | **0.296** |
| Exchange | 336 | **0.341** | **0.420** | 0.351 | 0.429 | **0.341** | **0.420** | 0.346 | 0.423 |
| | 720 | **0.854** | **0.696** | 0.872 | 0.705 | **0.854** | **0.696** | 0.861 | 0.700 |
| | Avg | **0.363** | **0.403** | 0.376 | 0.415 | **0.363** | **0.403** | 0.366 | 0.404 |

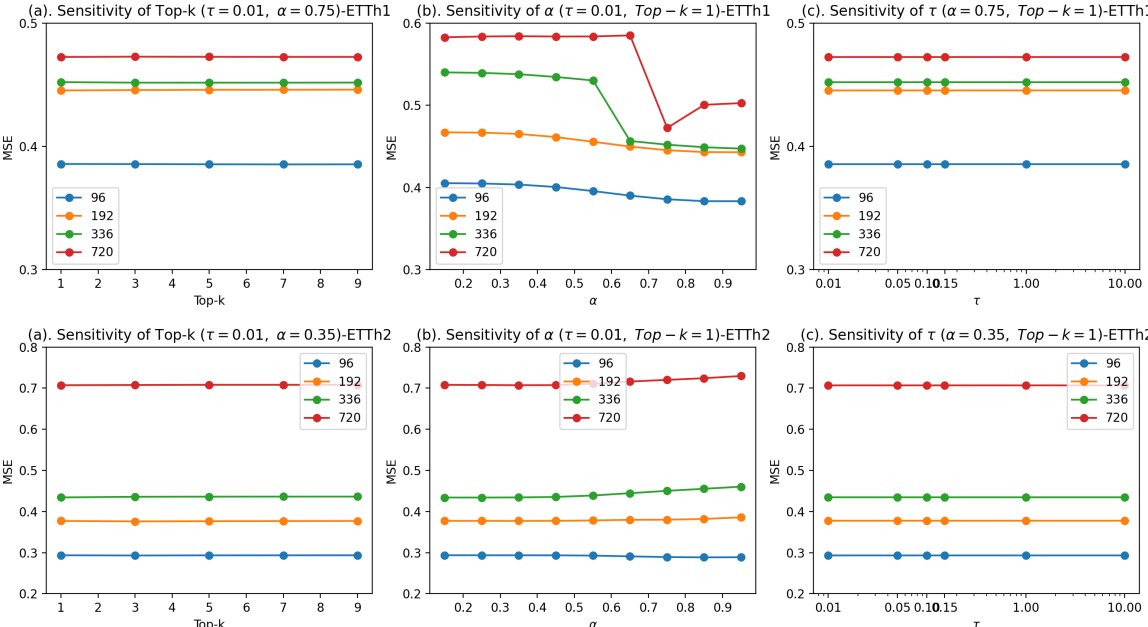

*Figure 5.* Sensitivity Analysis of KUP-BI Hyperparameters on DLinear across ETTh1 and ETTh2.

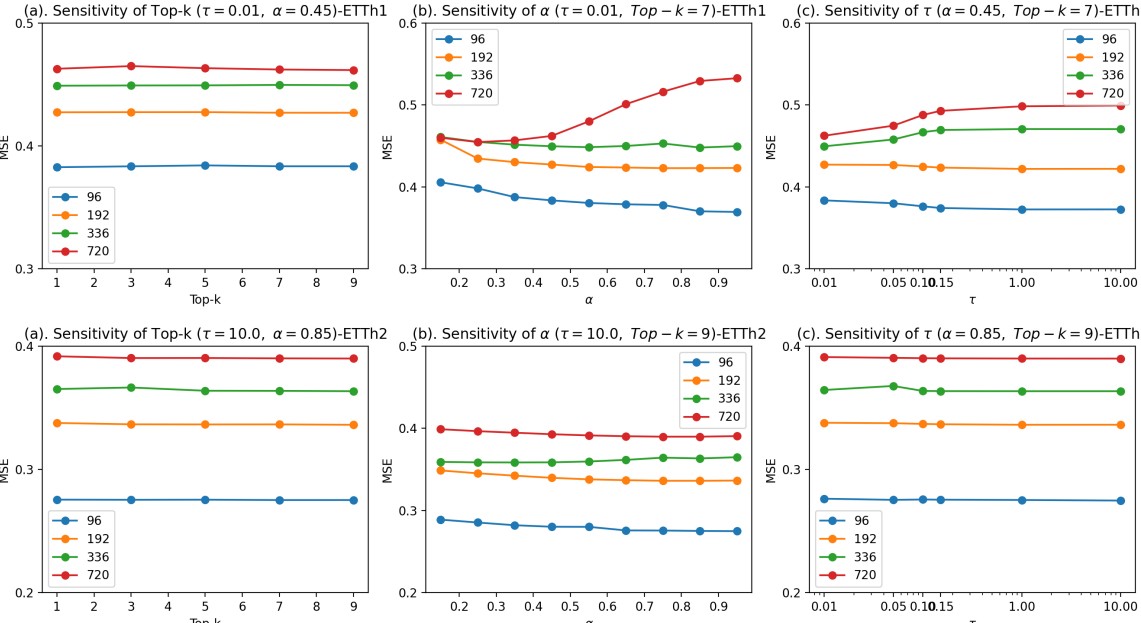

*Figure 6.* Sensitivity Analysis of KUP-BI Hyperparameters on xPatch across ETTh1 and ETTh2.

