# OpenReview forum: "Beyond Extrapolation: Knowledge Utilization Paradigm with Bidirectional Inspiration for Time Series Forecasting"
_ICML.cc/2026/Conference — ICML 2026 regular_

### Official Review · Reviewer_XVE5 · 2026-03-10

**Soundness:** 3
**Presentation:** 3
**Significance:** 4
**Originality:** 3
**Overall Recommendation:** 5
**Confidence:** 4

**Summary:**

In this work, the authors propose KUP-BI, a bidirectional reasoning-like time series modeling approach. Its core idea is to utilize estimated values of post-target continuations as supplementary information for the current input, thereby providing partial structural information about the prediction target.
KUP-BI achieves consistent gains over four different backbones across multiple datasets. Additionally, the authors provide extensive supplementary experimental results to support their work.
The authors present two independent approaches for estimating post-target continuations: retrieval-based and prediction-based methods, indicating this modeling framework holds potential for further extension.

**Compliance With Llm Reviewing Policy:**

Affirmed.

**Final Justification:**

The rebuttal addressed my concerns and reinforced my recommendation for acceptance.

**Key Questions For Authors:**

Q1:  As described above, I believe the authors should provide supplementary experiments. For instance, the extent of KUP-BI's improvement over the backbone model on large datasets;

Q2: One point I find curious: in the comparison with RAFT, were the authors aiming to demonstrate the relative strengths of different information sources, or the merits of two distinct methods? This requires clarification;

Q3: I noticed KUP-BI yields very small gains on some datasets. The authors can explain the reason for this?  If global tuning is disregarded, is such marginal improvement worth implementing? Does KUP-BI, like RAFT, exclude the most similar frame during training?

Q4: Based on my understanding, Equation (1) should represent element-wise operations, but the authors did not specify this.

**Limitations:**

yes

**Strengths And Weaknesses:**

Strengths:
+ KUP-BI is a novel framework, considering a completely different approach from previous retrieval-based work, namely, using post-target continuation as an auxiliary information source to enhance the predictive performance of the model；
+ The authors cleverly obtained estimates of post-target continuations for the current input from the training set through retrieval, thereby avoiding information leakage;
+ This concept is instantiated as a universally applicable method capable of boosting the accuracy of diverse prediction backbones in long-term forecasting tasks;
+ The paper's written is well.The authors detail their design process, acknowledge relevant limitations, and present a foundational assumption. Experimental procedures are well-documented.



Weaknesses:
+ I believe the method is effective, but the magnitude of these gains is not particularly striking. Can the authors explain what phenomenon might be causing this? The current results are not as impressive as one might expect;
+ Although the authors employed numerous datasets to validate their ideas, they did not provide experiments on large-scale datasets (e.g., weather or traffic). While this isn't a fatal flaw, it still leaves the impression that the work isn't entirely polished;
+ The authors did not validate the effectiveness of the proposed method on more advanced models (e.g., TSFM). What is the reason for this omission?

---

> ### Author Rebuttal · Authors · 2026-03-30
>
> Dear Reviewer XVE5,
>
> We sincerely thank the reviewer for the positive assessment and constructive suggestions. Since some of the Weaknesses and Questions are closely related, we organize our responses as W1/Q3, W2/Q1, W3, Q2, and Q4.
>
> ***[W1/Q3] Modest gains, their interpretation, and the training-time retrieval protocol***
>
> We agree that the gains are modest on some datasets. One practical reason is that, in the current experiments, we use a unified hyperparameter setting across all prediction horizons rather than tuning KUP-BI separately for each horizon/dataset. This design favors fairness and simplicity, but it can also limit the achievable gain in settings where the usefulness of the continuation prior is more sensitive to horizon or data characteristics. More broadly, KUP-BI is designed as a lightweight plug-in prior rather than a replacement for the backbone, so its goal is to provide small but stable, low-risk improvements across diverse settings.
>
> We still believe such gains are worthwhile in this context. The method adds only lightweight retrieval/fusion machinery, is backbone-agnostic, and can improve performance in many settings without requiring major architectural changes. In other words, its value lies not in producing dramatic gains in every setting, but in offering a simple and broadly applicable way to obtain modest but often consistent improvements.
>
> Regarding training-time retrieval, KUP-BI does not apply the same exclusion rule as RAFT during training. We will clarify the exact retrieval protocol in the revision to avoid ambiguity.
>
> ***[W2/Q1] Why not validate KUP-BI on larger-scale datasets such as Weather?***
>
> We thank the reviewer for this valuable suggestion. The core question here is whether post-target continuation can serve as a useful auxiliary prior for forecasting, and the same core mechanism is relevant on both smaller and larger datasets. Appendix Table 8 already provides partial evidence on this point: even when the retrieval signal is weak, introducing KUP-BI does not necessarily lead to overall degradation of the backbone. As an additional check, we also ran DLinear on the Weather dataset, where KUP-BI still yields a small positive improvement (MSE/MAE: 0.247/0.300 → 0.246/0.299). These results provide preliminary evidence that the proposed continuation-based design is not limited to the datasets in the main paper.
>
> ***[W3] Why no validation on TSFM-style models?***
>
> We thank the reviewer for this valuable suggestion. We agree that evaluating KUP-BI on stronger foundation-style models such as TSFM would be valuable. In the current paper, we focused on a diverse set of widely used forecasting backbones to validate the backbone-agnostic nature of KUP-BI under a relatively unified experimental setting. A fair comparison on TSFM-style models would ideally freeze the pretrained backbone and train only a small unified fusion/adaptation head, while keeping the retrieval/fusion protocol fixed, so that the contribution of KUP-BI can be isolated more clearly. We emphasize that this is a controlled evaluation choice rather than a universal requirement. In practice, existing TSFM-style retrieval frameworks already differ substantially in preprocessing, retrieval construction, input representation, and model-side priors, which makes a clean controlled comparison less straightforward within the scope of the current paper. We will clarify this point and consider TSFM-style validation as an important extension.
>
> ***[Q2] Clarification of the comparison with RAFT.***
>
> Thank you for pointing this out. In the original submission, the comparison with RAFT was mainly intended as a comparison between two retrieval-augmented designs, i.e., an overall method-level comparison. However, we agree that this setup does not fully isolate whether the observed difference comes from the auxiliary information source itself or from differences in the surrounding design.
>
> Following your suggestion, we therefore conducted a stricter controlled comparison under the same retrieval/fusion framework, where we only replaced the auxiliary signal from post-target continuations with the corresponding target segments while keeping the rest unchanged. Under this controlled setting, the continuation-based version performs better on average on ETTh1 (0.439/0.450 vs. 0.500/0.492), ETTh2 (0.453/0.458 vs. 0.500/0.487), ETTm2 (0.281/0.345 vs. 0.286/0.345), ILI (2.336/1.086 vs. 2.342/1.091), and Exchange (0.362/0.416 vs. 0.371/0.421). This additional experiment makes the intended message clearer: the key advantage comes from the continuation-style auxiliary information itself, rather than only from comparing two end-to-end methods.
>
> ***[Q4] Clarification of Eq.(1).***
> Yes. Eq. (1) is intended to be understood element-wise. We agree that this was not stated explicitly enough in the current version. We will clarify this in the revision and adjust the notation if needed to avoid ambiguity.

---

> > ### Author Rebuttal · Reviewer_XVE5 · 2026-04-02
> >
> > All concerns have been well addressed and I decide to keep the current score as 5 (accept)

---

### Official Review · Reviewer_gSt8 · 2026-03-10

**Soundness:** 2
**Presentation:** 3
**Significance:** 1
**Originality:** 2
**Overall Recommendation:** 4
**Confidence:** 3

**Summary:**

This paper proposes KUP-BI that goes beyond standard history-to-target prediction settings by introducing post-target continuation as a third stage. The key idea is to retrieve similar historical patterns from history data, summarize how their future trajectories continue to evolve, and use this continuation-style anxiulary signal to guide forecasting.

**Compliance With Llm Reviewing Policy:**

Affirmed.

**Final Justification:**

I have increased the score as the concerns are clear.

**Key Questions For Authors:**

1. My main question is whether the true post-target continuation is available or not? It seems that this method relies on an approximate proxy from retrieval.

**Limitations:**

same as weakness

**Strengths And Weaknesses:**

Strengths:
1. The paper proposes a new post-target continuation paradigm, which goes beyond standard history-to-target forecasting task.

2. It proposes a bidirectional inspiration paradigm, which combines forward forecasting with continuation-based auxiliary guidance, rather than relying only on one-way extrapolation.

3. The framework is plug-and-play, as KUP-BI can be integrated into multiple forecasting backbones instead of being tied to a single architecture.

4. Extensive empircial and visualization analyses are provided.

Weakness:

1. The absract section violates the 4-6 sentence requirement.

2. The motivation is interesting and novel, but its practical importance is not fully established. The paper does not clearly show that the proposed post-target continuation paradigm is an important topic in time-series forecasting.

3. The method seems useful for datasets with periodicity or stable long-term evolution. The usefulness of post-target continuation is likely dataset-dependent. We may need to investigate what scope of datasets are suitable for this method.

4. KUP-BI assumes that similar histories imply similar post-target continuation patterns, but this may fail under phase shifts or pattern changes.

5. The paper uses a train-only retrieval library for a continuation-style structural prior is an interesting twist on standard retrieval-augmented generation (RAG), but heavily relies on established correlation and gating techniques.

6.  The paper provides an intuitive design and empirical gains, but does not yet offer a strong theoretical analysis of when continuation-style auxiliary signals are helpful. Or we need to provide more demonstration to show whether these auxiliary signals are helpful.

---

> ### Author Rebuttal · Authors · 2026-03-30
>
> Dear Reviewer gSt8,
>
> We thank the reviewer for the constructive comments. Below we address each point and clarify the corresponding revisions. We hope these clarifications make the contribution, scope, and limitations of the work clearer.
>
> ***[K1]Is the true post-target continuation available?***
>
> It is available during training, but unavailable at inference time. KUP-BI therefore uses an estimated continuation proxy for training-inference consistency.
>
> A simple example may help. Let A denote the observed history, B the prediction target, and C the true post-target continuation after B. At inference time, only A is available, while both B and C lie in the future. KUP-BI therefore uses an estimated proxy C′ constructed from training data as an auxiliary structural prior.We use C′ rather than a target-aligned proxy B′ because B′ is more directly tied to the supervision target and may therefore be easier for the model to exploit as a shortcut during training, increasing the risk of weaker generalization. By contrast, C′ is intended to provide a weaker continuation prior rather than target-level guidance.
>
> ***[W1]The abstract section violates the 4-6 sentence requirement.***
>
> We will condense the abstract into a clearer 4-6 sentence version.
>
> ***[W2]Practical relevance of post-target continuation.***
>
> Post-target continuation is introduced not merely as a conceptual novelty, but to address a practical challenge in long-horizon forecasting: as the horizon grows, history alone may become insufficient to determine stable future evolution.
> This challenge is common in real-world forecasting. In particular, when the series exhibits periodicity or relatively stable long-term evolution, post-target continuation is more likely to contain reusable structure, making continuation-style priors practically useful when history alone is not sufficient.
>
> Its practical relevance is supported by our empirical findings. Across multiple datasets and backbones, the continuation-based design yields small but consistent average gains, especially in longer-horizon settings. Appendix Table 8 also suggests that even when retrieval is noisy, KUP-BI does not necessarily degrade overall backbone performance and can still provide modest gains in some settings, supporting its practical robustness.
>
> ***[W3]Applicability scope of KUP-BI across datasets.***
>
> We agree that the usefulness of post-target continuation is dataset-dependent. KUP-BI is most effective when post-target evolution contains reusable structure and the continuation proxy is reliable, and less suitable when future evolution is highly irregular, phase-shifted, or weakly transferable across similar histories. This is why it is particularly relevant for datasets with periodicity or relatively stable long-term evolution. Appendix Table 8 supports this view: the method tends to be more helpful when the continuation proxy is reliable and the task is not fully determined by observed history alone.
>
> ***[W4]Assumption 1 is difficult to hold in practice.***
>
> We agree that Assumption 1 is idealized. Under non-stationarity or phase misalignment, the retrieved continuation proxy can be less reliable, so gains may shrink or become negative. This assumption is not unique to our method; retrieval-based approaches often rely, either explicitly or implicitly, on a similar premise. Our goal is to explore this forecasting paradigm rather than retrieval design itself, so we keep retrieval relatively simple and leave more robust retrieval under distribution shift to future work.
>
> The current method already includes several mechanisms that improve robustness to inaccurate retrieval: offset-based matching, Top-k soft aggregation, clipping, and gated residual fusion all help reduce the influence of noisy auxiliary signals. As Appendix Table 8 shows, weak proxy quality usually leads to smaller or negative gains, but we do not observe consistent overall degradation.
>
> ***[W5] Use of established correlation and gating techniques.***
>
> This is intentional: our contribution lies in proposing and validating post-target continuation as a useful auxiliary prior, rather than in redesigning standard retrieval or fusion modules. Using simple modules helps isolate the value of the continuation signal itself while keeping the method lightweight and practical.
>
> ***[W6] When are continuation-style auxiliary signals helpful?***
>
> Appendix Table 8 provides initial empirical evidence on this point. In general, continuation-style auxiliary signals are most helpful when the continuation proxy is reliable and history alone is insufficient, especially when the forecasting horizon extends well beyond the input window.
>
> Poor proxy quality does not necessarily cause overall degradation. Our design, including gated fusion, reduces the influence of unreliable auxiliary signals. Thus, even when retrieval quality is weak, KUP-BI can often preserve overall backbone performance and may still provide gains in some settings.

---

> > ### Author Rebuttal · Reviewer_gSt8 · 2026-04-01
> >
> > For w4, I am still not convinced about the assumption as it is common we have phase shift/trend shift in the ts dataset.

---

> > > ### Author Response · Authors · 2026-04-02
> > >
> > > ***For w4, I am still not convinced about the assumption as it is common we have phase shift/trend shift in the ts dataset.***
> > >
> > > We agree with the reviewer that phase shift and trend shift are common in time-series data. ***In the revised manuscript, we will remove Assumption 1 and make it explicit that the current implementation can be affected by factors such as phase shifts or non-stationarity.***
> > >
> > > Our core contribution is to introduce a new forecasting paradigm based on post-target continuation, rather than to develop a new retrieval algorithm. Accordingly, at the implementation level, we follow the overall pipeline commonly used in existing retrieval-based forecasting methods, such as RAFT (Han et al., ICML 2025), while incorporating several robustness-oriented design choices. As a result, the current design can alleviate the effects of phase shifts or non-stationarity to some extent, but does not fully eliminate them.
> > >
> > > To further examine this concern, we extended KUP-BI with a simple phase-shift-aware retrieval refinement and evaluated it on ETTh1 with DLinear as the backbone. The refined version further improves the original KUP-BI on average (MSE/MAE: 0.439/0.450 to 0.437/0.447), with gains at 96, 192, and 336 horizons and slightly better MAE at 720.
> > > ***This preliminary result suggests that phase mismatch is indeed a real factor affecting retrieval quality, and that explicitly accounting for it can further improve forecasting accuracy.***
> > > We will also clarify in the revision that more explicit shift-aware retrieval or alignment is a promising direction for improving robustness under non-stationarity or phase/trend shifts.
> > >
> > > | ETTh1 (DLinear, seq_len=336) | +KUP-BI MSE | +KUP-BI MAE | Phase-aware MSE | Phase-aware MAE |
> > > |---|---:|---:|---:|---:|
> > > | 96  | 0.386 | 0.407 | 0.384 | 0.404 |
> > > | 192 | 0.445 | 0.451 | 0.444 | 0.450 |
> > > | 336 | 0.452 | 0.452 | 0.449 | 0.449 |
> > > | 720 | 0.472 | 0.489 | 0.472 | 0.487 |
> > > | **Avg** | **0.439** | **0.450** | **0.437** | **0.447** |

---

### Official Review · Reviewer_YdSb · 2026-03-12

**Soundness:** 2
**Presentation:** 1
**Significance:** 2
**Originality:** 2
**Overall Recommendation:** 3
**Confidence:** 5

**Summary:**

This paper proposes KUP-BI, a new time-series forecasting paradigm that leverages approximate post-target continuation as structural knowledge. By distilling continuation-style knowledge from a historical training library and fusing it with the input via a lightweight gating module, KUP-BI provides a structured inductive bias without introducing extra information. Experiments on six public datasets show consistent improvements over state-of-the-art models with minimal overhead.

**Compliance With Llm Reviewing Policy:**

Affirmed.

**Final Justification:**

Thank you to the authors for their detailed rebuttal and the additional experimental results they provided. The rebuttal and supplementary experiments have partially addressed my main concerns raised during the review stage, which led me to slightly increase my score for this paper. However, the paper still requires substantial revisions to strengthen its theoretical foundation and improve presentation.

**Key Questions For Authors:**

1. Why is the ratio-style representation between history and post-target continuation referred to as “knowledge”? Intuitively, it appears to align more closely with a retrieval pattern.
2. It is unclear why the ratio-style representation is adopted instead of using the raw post-target continuation. The authors are encouraged to clarify the motivation and provide more convincing evidence to justify this design choice.

**Limitations:**

Yes.

**Strengths And Weaknesses:**

Strengths:
1. The authors introduce a novel perspective for time series forecasting that breaks the limitations of relying solely on unidirectional extrapolation from history to target. By incorporating continuation-style auxiliary features, the approach successfully enables a bidirectional feature forecasting mechanism.
2. The authors propose an architecture-agnostic, plug-in general framework that consistently yields relative improvements in the predictive performance of mainstream time series models.
Weaknesses:
1. The motivation lacks rigorous theoretical justification. In the introduction, the authors claim that “target-level information can become an overly strong shortcut during training, which may weaken generalization.” However, no theoretical justification or supporting evidence is provided to substantiate this claim.
2. There are multiple ambiguous expressions and unclear presentations in the paper. For instance:
a) The authors propose two gated fusion methods, but fail to specify which one is actually employed in the experiments. Furthermore, there is no discussion or comparative analysis provided for these two methods.
b) The overall readability of the paper is suboptimal. Specific implementation details are prematurely introduced in the Introduction section. Additionally, the overly convoluted and lengthy descriptions accompanying the model figure make it challenging for readers to digest the architecture.
3. Assumption 1 is difficult to hold in practice. This assumption exhibits limitations in common time series scenarios, such as those involving distribution shifts, non-stationarity, and environmental disturbances.
4. The element-wise calculation in Equation (1) may exhibit limitations when dealing with temporal misalignments or complex evolutionary patterns.
5. The experimental setup in Section 4.3 is not well justified. The authors altered the parameters and architecture of RAFT in an attempt to isolate and prove that "post-target continuation" is superior to "target" for auxiliary forecasting. However, a more rigorous and convincing methodology would be to conduct a comparative study by simply replacing the values in the proposed model's retrieval library with targets. Consequently, the results presented in Table 2 are not entirely persuasive.
6. The ablation study is not well designed. The authors primarily conducted ablations at the parameter level rather than on the key architectural components.

---

> ### Author Rebuttal · Authors · 2026-03-30
>
> Dear Reviewer YdSb,
>
> Thank you for the careful assessment. We clarify the motivation, fusion setting, scope of Assumption 1, limitation of Eq.(1), target-vs-continuation comparison, and ablations. We hope these revisions address your concerns.
>
> ***[W1]The motivation lacks rigorous support.***
>
> We agree that the original statement would be too strong if interpreted as a formal theoretical claim. Our point is more modest and empirical: target-aligned auxiliary signals may act as easier shortcuts during training, reducing the need to learn more transferable patterns and potentially weakening generalization.
>
> To support this motivation, we conducted a controlled comparison under the same retrieval/fusion framework, replacing post-target continuations with target segments while keeping the rest unchanged. The continuation-based version performs better on average; for example, on ETTh1 it achieves 0.439/0.450 vs. 0.500/0.492, and on Exchange 0.362/0.416 vs. 0.371/0.421. We will report the full results in the revision.
>
> ***[W2]The paper contains several ambiguities and presentation issues.***
>
> In our experiments, the static gate is the default configuration, while the dynamic gate is only an optional variant. We will make this explicit in the revision by annotating the fusion configuration in the experiments/tables and briefly discussing the two fusion forms. We will also move implementation details out of the Introduction and simplify the figure-related text.
>
> ***[W3]Assumption 1 is difficult to hold in practice.***
>
> We agree that Assumption 1 is idealized. Under non-stationarity or phase misalignment, the retrieved continuation proxy can be less reliable, so gains may shrink or become negative. This assumption is not unique to our method; retrieval-based approaches often rely, either explicitly or implicitly, on a similar premise. Our goal is to explore the effectiveness of a new forecasting paradigm rather than retrieval design itself, so we keep retrieval relatively simple. We agree that handling stronger non-stationarity or phase shifts is important, and leave more robust retrieval under distribution shift to future work.
>
> The current method already includes several mechanisms that improve robustness to inaccurate retrieval: offset-based matching, Top-k soft aggregation, clipping, and gated residual fusion all help reduce the influence of noisy auxiliary signals. As Appendix Table 8 shows, weak proxy quality usually leads to smaller or negative gains, but we do not observe consistent overall degradation.
>
> ***[W4]Eq.(1) may not handle temporal misalignment well.***
>
> We agree with this concern. As a lightweight design, Eq.(1) does not explicitly model alignment. However, KUP-BI is not based on Eq.(1) alone: last-step offsetting, Top-k soft aggregation, quantile-tanh clipping, and gated residual fusion reduce sensitivity to mismatched retrieved continuations. Appendix Table 8 further shows that even when proxy quality is weak, KUP-BI does not necessarily degrade overall backbone performance.
>
> ***[W5]The setup in Section 4.3 is not sufficiently justified.***
>
> Following your suggestion, we replaced post-target continuations with target segments under the same retrieval/fusion framework. The results in [W1] support the continuation-based design, and we will clarify this comparison in the revision.
>
> ***[W6]The ablation study is not well designed.***
>
> We agree that the ablation presentation was not clear enough. Section 4.4 already includes controlled variants on fusion, retrieval, and continuation construction/representation, but these analyses were not highlighted clearly enough. In the revision, we will reorganize them into a clearer ablation structure and add the ratio-vs-residual comparison to make the component-level analysis more explicit.
>
> ***[Q1]Why refer to the ratio-style representation as “knowledge”?***
>
> To clarify, we do not regard the ratio-style representation itself as “knowledge.” What we refer to as “knowledge” is the post-target continuation beyond the target segment.
> We use the ratio-style representation as one way to represent this estimated knowledge.
> Our intention is not to bind this idea to retrieval itself, which is also reflected in the fact that we consider both retrieval-based and prediction-based continuation construction.
>
> ***[Q2]Why use the ratio representation instead of the raw continuation?***
>
> The ratio-style representation is a simple, stable descriptor of how the post-target continuation evolves relative to history. Compared with raw continuation, it is more scale-aware and less sensitive to level mismatch in the retrieved continuations. We support this choice in two ways. First, Section 4.4 already includes the Direct (raw) Continuation variant, where the ratio-based version performs better overall. Second, under the same retrieval/fusion setting, replacing ratio with an additive residual also gives worse results, for example on ETTh1 (MSE/MAE), 0.439/0.450 vs. 0.445/0.455.

---

> > ### Author Rebuttal · Reviewer_YdSb · 2026-04-06
> >
> > 1. After reading the other reviewers’ comments, I find that W3 remains an important concern shared by multiple reviewers, including myself.
> > 2. In the authors’ response to Q1, the statement *“What we refer to as ‘knowledge’ is the post-target continuation beyond the target segment”* still appears difficult to justify conceptually.
> > 3. The overall presentation and writing could be further improved for clarity. While the authors have indicated plans for substantial revisions (including W1, W2, W5, and W6), the number and extent of these issues suggest that significant effort would be required to bring the paper to the expected standard.
> > 4. The motivation of the proposed method is primarily supported by empirical results; additional theoretical or conceptual insights could further strengthen the work.

---

> > > ### Author Response · Authors · 2026-04-07
> > >
> > > Dear Reviewer YdSb,
> > >
> > > Thank you for the follow-up comments. We respectfully hope the reviewer will consider the following clarifications and planned revisions in a balanced manner, and assess the paper in light of its actual scope and contribution.
> > >
> > > ***Q1: After reading the other reviewers’ comments, I find that W3 remains an important concern shared by multiple reviewers, including myself.***
> > >
> > > Although this issue was also raised by the other two reviewers, it did not prevent them from giving overall positive assessments of the work.
> > >
> > > For W3, we will remove Assumption 1 from the revised manuscript and make it explicit that the current implementation can be affected by factors such as phase shifts or non-stationarity.
> > > Our core contribution is to introduce a new forecasting paradigm based on post-target continuation, rather than to develop a new retrieval algorithm. Accordingly, at the implementation level, we follow the overall pipeline commonly used in existing retrieval-based forecasting methods, such as RAFT (Han et al., ICML 2025). As a result, while the current design can alleviate the effects of phase shifts or non-stationarity to some extent, it does not fully eliminate them. More broadly, the effect of phase shifts or non-stationarity on retrieval quality is a challenging problem that will likely require continued progress in future work on retrieval robustness, alignment, and shift-aware matching.
> > >
> > >
> > > ***Q2: In the authors' response to Q1, the statement “What we refer to as ‘knowledge’ is the post-target continuation beyond the target segment” still appears difficult to justify conceptually.***
> > >
> > > We thank the reviewer for this clarification. We agree that “knowledge” may be somewhat broad in the current wording. Our intent is not to treat the post-target continuation itself as formal “knowledge,” but rather as reusable continuation information or an auxiliary continuation prior. We will revise the terminology accordingly to avoid overstating the conceptual meaning.
> > >
> > > ***Q3: The overall presentation and writing could be further improved for clarity. While the authors have indicated plans for substantial revisions (including W1, W2, W5, and W6), the number and extent of these issues suggest that significant effort would be required to bring the paper to the expected standard.***
> > >
> > > We respectfully feel that describing the manuscript as requiring substantial revision may somewhat overstate the extent of the changes needed.
> > >
> > > First, W5 mainly requests an additional experiment, which has already been completed. In addition, the structure-related ablations relevant to W6 were already included in Section 4.4 of the original manuscript. Moreover, W1 and W2 mainly concern terminology and presentation rather than substantial methodological issues.
> > >
> > > For this reason, the required revisions are primarily about clarification, wording, and presentation polish, rather than major methodological changes.
> > >
> > >
> > > ***Q4: The motivation of the proposed method is primarily supported by empirical results; additional theoretical or conceptual insights could further strengthen the work.***
> > >
> > > Indeed, the motivation in the current version is mainly supported by empirical results.
> > >
> > > Conceptually, the key idea is that, in long-horizon forecasting, the historical input alone may be insufficient to determine stable future evolution. In such cases, post-target continuation can serve as an auxiliary continuation prior that provides complementary information beyond the observed history.
> > >
> > > This is the main conceptual motivation behind our method, and we will make it more explicit in the revision.

---

### Official Review · Reviewer_RqyZ · 2026-03-12

**Soundness:** 3
**Presentation:** 4
**Significance:** 2
**Originality:** 3
**Overall Recommendation:** 4
**Confidence:** 4

**Summary:**

This paper proposes KUP-BI, a plug-in forecasting paradigm that augments the usual history-to-target prediction path with a continuation-style auxiliary stream built from a train-only retrieval library of “history–target–post-target continuation” chains. The key idea is to leverage patterns observed in training data beyond the prediction window. Specifically, they construct a retrieval library consisting of triples of historical segments, their corresponding targets, and the subsequent post-target continuations. When inference, the model retrieves historical segments similar to the current input sequence using channel-wise Pearson correlation. Then the retrieved samples provide continuation descriptors, represented as ratio-style transformations relative to the prediction target. These descriptors are aggregated through a softmax-weighted scheme to form an auxiliary continuation signal. The resulting continuation estimate is then converted into an auxiliary sequence and integrated with the backbone forecasting model through a gated residual fusion module. The overall framework is designed as a lightweight plug-in and can be applied to different forecasting architectures without modifying their core structure.

Experiments are on six benchmark time-series datasets using four forecasting backbones (including linear and transformer-based models). The results show consistent performance improvements, particularly for longer prediction horizons, while introducing minimal additional computational overhead. The authors also compare the proposed continuation-based retrieval strategy with a target-based retrieval baseline and conduct several ablation studies to evaluate design choices. Limitations discussed by the authors include sensitivity to retrieval design, possible phase-shift mismatches, and the lack of a formal theoretical justification for the continuation proxy.

**Compliance With Llm Reviewing Policy:**

Affirmed.

**Key Questions For Authors:**

1. Justification for the ratio-style operator.
The continuation signal is constructed using a ratio-style operator. While the authors argue that this keeps the module non-parametric, it is unclear why this multiplicative formulation was chosen over simpler alternatives such as additive residuals, normalized differences, or learned embeddings. Could the authors clarify the motivation for this design choice, or indicate whether alternative formulations were tested during development?

2. Robustness of the retrieval assumption under non-stationarity.
The model relies on the conditional stationarity of continuation patterns (Assumption 1). However, the authors acknowledge that the current retrieval strategy does not explicitly handle phase shifts. Since datasets such as Exchange may contain such misalignments, how does the model behave when the retrieved continuation proxy is inaccurate or noisy? It would be helpful to see analysis of how sensitive the method is to such distribution shifts.

3. Analysis of cases where the method does not improve performance.
While KUP-BI shows consistent average gains, Table 1 suggests that the improvements are relatively small in some settings (e.g., ETTh1 with DLinear). The paper would be strengthened by an analysis of failure cases—for example, identifying scenarios (such as series with low autocorrelation) where the continuation proxy may degrade performance or introduce noisy auxiliary signals.

4. Qualitative analysis of the retrieval mechanism.
The concept of “Bidirectional Inspiration” is a central idea in the paper. However, the results are presented almost exclusively through quantitative tables. To better understand the mechanism, could the authors provide qualitative visualizations? Specifically, showing examples where retrieved training trajectories influence or correct the backbone’s predictions would help illustrate how the continuation proxy improves predictions.

**Limitations:**

Yes.

**Strengths And Weaknesses:**

Strengths:
1. Addresses an important and challenging problem in time-series forecasting.
Long-horizon forecasting remains difficult due to error accumulation and trend drift over extended prediction horizons. Improving stability and reliability at longer horizons is still an important research objective. The paper directly targets this issue, and the reported improvements become more visible as the prediction horizon increases. This aligns well with the paper’s motivation and suggests the approach can be helpful for long-horizon forecasting.

2.  Conceptually interesting and moderately novel methodological idea.
The paper introduces a retrieval-based forecasting approach that leverages post-target continuation patterns rather than retrieving target segments directly. This reframes retrieval as providing a prior over possible future dynamics and differs from existing target-retrieval strategies. The idea itself is fairly simple and is implemented as a lightweight plug-in module consisting of retrieval, continuation-ratio construction, aggregation, auxiliary sequence generation, and gated fusion. Since the method only adds a small auxiliary component to the model, it is relatively easy to understand and could potentially be integrated with a wide range of forecasting architectures.

3. Reasonably comprehensive experimental evaluation.
The method is evaluated across four forecasting backbones and six benchmark datasets, suggesting that the idea is not limited to a single architecture or dataset. The evaluation includes both simple linear models (e.g., DLinear) and more complex transformer-style architectures (e.g., PatchTST and TimesNet), demonstrating compatibility with different modeling paradigms. The inclusion of a retrieval-based baseline also helps clarify how continuation-based retrieval differs from standard target-based retrieval approaches.

4. Clear structure and reasonable robustness analysis.
The paper presents the method through a well-organized pipeline, which makes the overall design easy to follow. The authors also include ablation studies examining several design components, such as retrieval size, temperature scaling, continuation scaling, and the fusion module. In addition, the paper clarifies that the continuation library is constructed using training data only, avoiding the use of ground-truth future values during inference. The authors also discuss several limitations of the approach, including potential phase-shift issues and the heuristic nature of some design choices, which helps improve transparency.

Weaknesses:

1. Sensitivity of the Retrieval Assumption to Non-Stationarity
The paper assumes that similar historical windows imply similar post-target continuations, which is sometimes fragile in real-world time series. Some datasets used in the paper (e.g., ETT and Exchange) may exhibit regime shifts, non-stationarity, or phase misalignment. The choice of channel-wise Pearson correlation as a similarity metric is particularly susceptible to these shifts. The paper does not analyze how the model behaves when historical patterns look similar, but the underlying dynamics have changed, which could lead the model to inject misleading auxiliary signals.

2. Marginal Significance and Lack of Failure Analysis
Although the method generally improves over baseline models, the improvements in some settings appear relatively modest. For example, in Table 1 several backbone–dataset combinations show only small reductions in error compared with the original models. Similarly, the comparison with RAFT in Table 2 does not consistently show clear advantages across all settings. The paper would be significantly strengthened by a failure case analysis. Since the method relies on external "inspiration" from a library, it is critical to know when this auxiliary signal is detrimental (e.g., in highly irregular or low-autocorrelation series). Without knowing where the "break point" is, it remains somewhat unclear when the auxiliary continuation signal provides the most benefit.

3. Limited theoretical justification
While the paper provides a toy model to illustrate the idea, the analysis mainly considers a simplified linear setting and does not fully justify the ratio-style operator or the gated fusion mechanism used in the deep-learning backbones. The paper also does not clearly explain why the proposed ratio-based transformation should work better than other possible representations (e.g., additive residuals or learned embeddings). In addition, the use of a manually chosen epsilon term to avoid division by small values suggests the formulation may not be enough robust mathematically. Without a clearer objective or theoretical analysis, it is difficult to understand under what conditions this continuation representation is expected to generalize well.

4. Practical complexity
The paper presents KUP-BI as a lightweight plug-in module, but the actual implementation introduces additional complexity. The retrieval library must be stored and searched during inference, and the cost of this process will grow with the size of the training data. The paper does not provide detailed analysis of how memory usage and inference time scale for large datasets. In addition, the experiments indicate that joint tuning with the backbone model improves performance, which suggests the method may require some tuning rather than functioning as a purely plug-and-play module.

5. Limited qualitative analysis
Retrieval-based methods can often provide useful interpretability, since one can inspect the retrieved examples that influence the prediction. However, the paper mainly reports quantitative results and does not provide qualitative visualizations of the retrieved continuations. Showing examples of retrieved trajectories and how they affect the final forecast could help illustrate how the continuation signal influences the prediction.

---

> ### Author Rebuttal · Authors · 2026-03-30
>
> Dear Reviewer RqyZ,
>
> Thank you for the careful reading and constructive comments. Since some Weaknesses and Questions overlap, we reply jointly as W1/Q2, W2/Q3, W3/Q1, W4, and W5/Q4.
>
> ***[W1/Q2] Robustness to non-stationarity and noisy retrieval***
>
> We agree that Assumption 1 is idealized. Under non-stationarity or phase misalignment, the retrieved continuation proxy can be less reliable, so gains may shrink or become negative. This assumption is not unique to our method; retrieval-based approaches often rely, either explicitly or implicitly, on a similar premise. Our goal is to explore this forecasting paradigm rather than retrieval design itself, so we keep retrieval relatively simple. We agree that handling stronger non-stationarity or phase shifts is important, and leave more robust retrieval under distribution shift to future work.
>
> We would also like to point out that the current method already includes several mechanisms that improve robustness to inaccurate retrieval: offset-based matching, Top-k soft aggregation, clipping, and gated residual fusion all help reduce the influence of noisy auxiliary signals. As Appendix Table 8 shows, weak proxy quality usually leads to smaller or negative gains, but we do not observe consistent overall degradation.
>
> ***[W2/Q3] Failure Cases and Applicability Boundaries of KUP-BI***
>
> We agree that failure-case analysis is important. Our current evidence suggests that it is most effective when the retrieved continuation proxy is reliable and the forecasting horizon is long enough that history alone becomes insufficient. Otherwise, gains can be small or negative. Appendix Table 8 supports this: weak proxy quality is usually associated with limited or negative gains, while strong proxy quality alone does not guarantee improvement. For example, on Exchange, the proxy remains relatively strong, but gains are still negative at 96/192/336 and become clearly positive only at 720, suggesting that with the input window fixed at 336, history may already be sufficient for shorter horizons, while the continuation prior becomes more useful for longer-horizon extrapolation. We will make these applicability boundaries and failure-prone cases more explicit in the revision.
>
> ***[W3/Q1] Design Justification for Ratio Representation and Gated Fusion***
>
> (1)	Ratio Representation: The ratio-style operator is a simple stabilized descriptor of how the post-target continuation evolves relative to its preceding history. Compared with an additive residual, it is more scale-aware, since the same absolute difference can have very different meanings across channels or samples with different magnitudes.  Unlike a symmetric normalized difference such as (A−B)/(A+B), it is explicitly history-anchored: history is the reference, and continuation is described relative to it. This better matches our goal of characterizing post-target evolution conditioned on preceding history. We also tested alternative formulations under the same retrieval/fusion setting. For example, on ETTh1, ratio achieves 0.439/0.450 versus 0.445/0.455 for residual and 0.480/0.480 for normalized difference. More complete results will be included in the revision.
>
> (2)	Gated fusion. We do not claim theoretical optimality; it is a simple and widely used robustness-oriented fusion design that allows the auxiliary branch to modulate rather than replace the main stream when proxy quality is imperfect.
>
> (3)	Epsilon. The ϵ term is a standard numerical stabilizer for ratio normalization. Robustness does not rely on it alone, since we also use quantile-tanh clipping and gated residual fusion.
>
> ***[W4] Practical complexity***
>
> We agree that “lightweight plug-in” should be stated more precisely. The online stage does not search the full training set. Retrieval is precomputed offline: for each query window, we cache a fixed Top-10 shortlist. During training/inference, the model only re-ranks this shortlist and aggregates the Top-k retrieved ratio vectors with softmax weighting, followed by lightweight fusion. Thus, online cost is bounded by a small constant-size candidate set rather than the full training set. On ETTh1, varying retrieval library size from 25% to 100% substantially increases cache size (e.g., for pred_len=96, from 25 MB to 282.55 MB), while inference time stays around 0.60–0.70 ms/batch. Thus, the main scaling cost is offline storage, while online overhead remains small. We will also clarify that “plug-in” means low integration cost rather than zero tuning cost.
>
> ***[W5/Q4] Qualitative analysis of the retrieval mechanism***
>
> We agree that qualitative analysis would make the influence of the continuation signal clearer. We have conducted such visualizations, and in the revision we will add examples showing the retrieved continuations, the estimated continuation used as auxiliary input, and the resulting forecasts of the backbone and KUP-BI against the ground truth.

---

### Decision · Program_Chairs · 2026-04-30

**Decision:**

Accept (regular)

**Comment:**

This paper proposes KUP-BI, a forecasting framework that augments standard history-to-target prediction with an auxiliary continuation-style signal derived from training data, with the goal of improving long-horizon forecasting. Reviewers generally agreed that this is an interesting and novel perspective, and they found the method lightweight, broadly compatible with different backbones, and empirically promising across multiple datasets.

The main strengths identified by reviewers were the continuation-based formulation itself, the consistent if modest empirical gains across backbones, and the fact that the method appears to provide a useful auxiliary prior without substantial additional overhead. At the same time, reviewers raised concerns about the conceptual and theoretical justification, especially the assumptions underlying retrieval under non-stationarity or phase shift, as well as the fact that gains are sometimes modest and may be dataset-dependent. Some reviewers also noted presentation issues and wanted clearer component-level justification and failure-case analysis.

The rebuttal addressed a number of these concerns with clarifications, additional controlled comparisons, and further analysis, and I have taken those responses into account. While some limitations remain, especially around robustness under shift and the strength of the conceptual framing, the overall discussion supports the view that the paper introduces a useful and sufficiently validated new forecasting idea. On balance, I recommend acceptance.